# Persisting volcanic ash particles impact stratospheric SO$_2$ lifetime and aerosol optical properties

Yunqian Zhu [1✉], Owen B. Toon[1,2], Eric J. Jensen[3], Charles G. Bardeen[3], Michael J. Mills [3], Margaret A. Tolbert[4], Pengfei Yu[5,7] & Sarah Woods[6]

Volcanic ash is often neglected in climate simulations because ash particles are assumed to have a short atmospheric lifetime, and to not participate in sulfur chemistry. After the Mt. Kelut eruption in 2014, stratospheric ash-rich aerosols were observed for months. Here we show that the persistence of super-micron ash is consistent with a density near 0.5 g cm$^{-3}$, close to pumice. Ash-rich particles dominate the volcanic cloud optical properties for the first 60 days. We also find that the initial SO$_2$ lifetime is determined by SO$_2$ uptake on ash, rather than by reaction with OH as commonly assumed. About 43% more volcanic sulfur is removed from the stratosphere in 2 months with the SO$_2$ heterogeneous chemistry on ash particles than without. This research suggests the need for re-evaluation of factors controlling SO$_2$ lifetime in climate model simulations, and of the impact of volcanic ash on stratospheric chemistry and radiation.

[1] Laboratory for Atmospheric and Space Physics, University of Colorado, Boulder, CO 80303, USA. [2] Department of Atmospheric and Oceanic Sciences, University of Colorado, Boulder, CO 80302, USA. [3] Atmospheric Chemistry Observations and Modeling Laboratory, National Center for Atmospheric Research, Boulder, CO 80301, USA. [4] Cooperative Institute for Research in Environmental Sciences, University of Colorado, Boulder, CO 80309, USA. [5] Earth System Research Laboratory, National Oceanic and Atmospheric Administration, Boulder, CO 80305, USA. [6] Stratton Park Engineering Company, Inc, Boulder, CO 80301, USA. [7] Present address: Institute for Environmental and Climate Research, Jinan University, Guangzhou, China. ✉email: yunqian.zhu@colorado.edu

Stratospheric volcanic aerosols have altered climate throughout Earth's history. The major constituents of volcanic aerosols are liquid sulfate ($SO_4^{2-}$ in solution with water) originating from sulfur dioxide ($SO_2$) injections, and volcanic rocks <2 mm in diameter, referred to as ash. It is well known that the sulfate aerosol (a.k.a. sulfuric acid aerosol) scatters incoming solar radiation back to space and cools the surface globally, while its absorption of solar and infrared light heats the stratosphere. Sulfate aerosols also provide surfaces to activate halogen species, enhancing ozone depletion. Volcanic ash is usually neglected in simulations of changes in the climate and stratospheric chemistry because large, high-density (over 2 g cm$^{-3}$) particles would be short lived[1].

However, observations indicate that ash can linger for long periods of time in the stratosphere. The stratospheric volcanic aerosol layer from the tropical Mt. Kelut eruption of 13 February, 2014, with Volcanic Explosivity Index (VEI) of 4, was observed for more than three months by the Cloud-Aerosol Lidar and Infrared Pathfinder Satellite Observation (CALIPSO)[2]. Vernier et al.[2] suggested that sub-micron volcanic ash particles were partly responsible for the aerosol layer. The Airborne Tropical TRopopause EXperiment (ATTREX) aircraft observed super-micron particles in the lower stratosphere near Guam 3–4 weeks after the eruption that was very likely volcanic ash[3]. Kelut's cloud is not a unique example of persisting ash-containing particles. Six to eight months after the 1991 Mt. Pinatubo eruption (VEI = 6), particles with mean radii near 0.8 μm composed of minerals coated with sulfate were observed in the Arctic lowermost stratosphere, increasing the particulate surface area up to 50-fold[4]. Similarly, volcanic ash was observed in the stratosphere for a year after the 1963 Mt. Agung eruption[5] (VEI = 5) and for a year after the 1982 El Chichon eruption[6] (VEI = 5).

The Mt. Kelut eruption is one of the few since those of Mt. Agung[5] and Mt. St. Helens[7] in 1980 for which ash size distributions were observed soon after the eruptions. Some of the volcanic ash particles are relatively large, have a significant fall velocity, and can remove the sulfate attached to them from the stratosphere. Some of the smaller ash particles contribute to the longer-lasting stratospheric volcanic aerosols, which can affect the radiation and chemistry for a longer time. Current simulations often assume ash has a density of ~ 2.3 g cm$^{-3}$, equivalent to volcanic glass[8]. Such dense particles of 10 μm radius have fall velocities near 4 cm s$^{-1}$ at 20 km, and fall times from 20 to 16 km of about 1 day. The lifetime of 1-μm particles of the same density is about 50 days, or slightly more than 200 days if the density is 0.5 g cm$^{-3}$. Non-spherical shapes can also reduce fall velocity.

Here, we compare simulations using a sectional aerosol-climate model (detailed in "Methods") with satellite and aircraft observations to investigate the physical processes that explain the persisting ash particles following the Kelut eruption. We analyze the ash physical properties (particle density, particle size distribution, etc.) and determine the temporal contribution of volcanic ash to the stratospheric volcanic aerosol layer. We find that the $SO_2$ lifetime is not controlled only by hydroxyl radical (•OH) as commonly assumed, but also by heterogeneous reactions on ash, a potential concern for climate and geoengineering simulations. Finally, we show that there is significant removal of sulfur (S) on falling ash. Such removal has been observed in the troposphere near volcanic vents possibly due to adsorption of S gases on the ash[9–14]. However, $SO_2$ removal via heterogeneous reactions on ash in the stratosphere has been ignored in climate models, even though it is recognized in laboratory studies[15–18].

## Results

**$SO_2$ lifetime.** Correctly constraining the volcanic $SO_2$ emission and lifetime, as well as the physical proximity of S gases and ash particles, is important to predict the interaction between S species and ash particles. Large ash particles have large surface areas to uptake the vapor through condensational growth or heterogeneous reactions, but at the same time, they quickly fall out of the stratosphere. Ash can interact with $H_2SO_4$ gas and aerosol through several well-known microphysical processes (Methods, Fig. 1): $H_2SO_4$ gas heterogeneously nucleates on ash particles and then grows; pure sulfate aerosol particles form by homogeneous nucleation and can then coagulate with ash particles and mixed ash/sulfate particles. The longer it takes $SO_2$ to be converted to $H_2SO_4$, the less $H_2SO_4$ vapor is available to condense onto the ash, and the fewer sulfate particles are available to coagulate on ash particles before the ash particles fall out of the stratosphere. Heterogeneous uptake of $SO_2$ on ash can remove $SO_2$ directly without requiring $SO_2$ conversion to $H_2SO_4$ vapor (Methods, Fig. 1).

The $SO_2$ oxidation process is very complicated inside a volcanic plume and cloud. Generally, in the stratosphere, it has been thought that the $SO_2$ lifetime is determined by its reaction rate with OH. In turn, the oxidation of $SO_2$ reduces the OH abundance. Therefore, OH chemistry must be included in any simulation of volcanic plumes and clouds. The factors that impact the $SO_2$/OH reaction rate include the injection of $H_2O$ from the volcanic eruption, which can increase the OH concentration; the spreading of the volcanic clouds, which dilutes the $SO_2$ concentration so OH is less impacted by the reaction with $SO_2$; as well as light scattering by ash, sulfate, and possibly ice, which increases the path length of light through the atmosphere thereby changing photolysis rates that control the OH production rate[19].

Carn et al.[20] discovered a correlation between the altitude of $SO_2$ injection and the lifetime of $SO_2$ and possibly a related dependence of $SO_2$ lifetime on the amount of injected $SO_2$ and the latitude of injection. They found that large injections, such as that of Mt. Pinatubo, in the altitude range above 20 km have $SO_2$ lifetimes near 30 days, while smaller injections near the tropopause result in lifetimes of less than a week.

To explore the influence of various factors that might alter the $SO_2$ oxidation rate in our model, we show in Fig. 2 several simulations of $SO_2$ evolution (lines), as well as various spacecraft observations (symbols). The dashed lines in Fig. 2 represent the $SO_2$ burden from the simulations. As the volcanic cloud spreads, the $SO_2$ column is reduced due to wind shear until some columns have so little $SO_2$ that it is below the observational detection limit, but is not actually lost. Therefore, some apparent $SO_2$ loss is actually $SO_2$ spreading into the background. The solid lines in Fig. 2 represent the simulated $SO_2$ burden only counting the grid cells with $SO_2$ above the background noise level of the Ozone Monitoring Instrument (OMI) data (reasons detailed in Supplementary Note 1).

Mills et al.[21] assumed an injection of 0.3 Tg of $SO_2$ from Kelut, and computed an $SO_2$ lifetime of 26 days. However, the satellite observations in Fig. 2 report that the $SO_2$ lifetime after the Kelut eruption is about a week. Considering the $SO_2$ merging into the background with signals below instrumental detection limits, our simulations suggest the actual $SO_2$ lifetime is ~2 weeks. The Reference case in Fig. 2 and Table 1 shows that reducing the injected amount of $SO_2$ from 0.3 Tg to 0.2 Tg does not impact the $SO_2$ lifetime found by Mills et al.[21]. The Base case explores the effect on the $SO_2$ lifetime by altering three factors to be closer to the Microwave Limb Sounder (MLS) observed $SO_2$ and $H_2O$. First, we used an initial injection into a 10° latitude × 2° longitude band so that we could pick up enough wind shear to reproduce the spreading patterns and concentrations observed by the OMI satellite (Supplementary Note 2; Supplementary, Figs. 1–3). Second, we used a lower $SO_2$ injection height than suggest by Mills et al.[21] based on MLS observations (Table 1). Third, we used

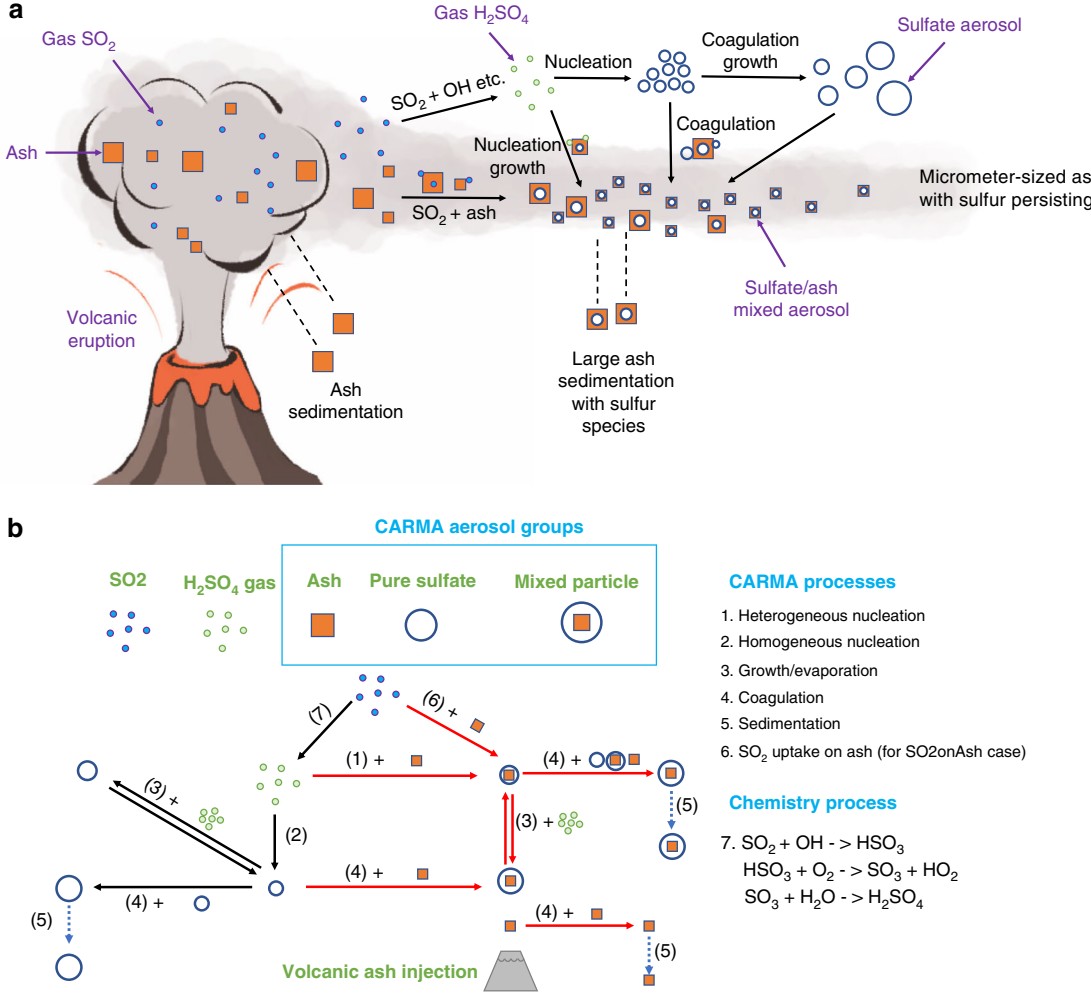

**Fig. 1 A schematic diagram and the volcanic ash/sulfate module in the model. a** A schematic diagram of the chemistry and microphysical processes for this work; **b** The three aerosol groups and the main microphysical processes involved in the volcanic ash/sulfate module in the CARMA model. The red arrows indicate the processes added to the model for this study relative to English et al.[42].

a water injection of ~0.26 Tg which is near the upper limit allowed by MLS (Table 1). We included these three factors together because their effects are each small, in total they shorten the lifetime by 4 days relative to the Mills et al.[20] lifetime of 26 days. With three tests tracking the three factors individually, we found the larger geographic injection area for $SO_2$ contributes 2 days to lifetime reduction, the lower injection height contributes 2 days, and the observed $H_2O$ injection causes no significant reduction.

Figure 2 shows that none of the simulations just described reproduce the satellite observations of $SO_2$. Supplementary Fig. 4 illustrates the OH field 1 day after the eruption from the Base case. As may be seen OH in the volcanic cloud is depleted by ~1 pptv relative to surrounding air. Basically, OH is converted to $HO_2$, so that the ratio of OH to $HO_2$ is changed. The slow down of the $SO_2$/OH reaction due to depletion of OH has been suggested in models[22] and seen in oxygen isotopic data during historical moderate and large eruptions[23–25]. One possible way to increase OH, and thereby reduce the $SO_2$ lifetime is to assume that a large injection of $H_2O$ occurred. In order to fit the observed $SO_2$ trend, we need to inject 26 Tg of $H_2O$ (the LrgH2O case in Table 1, not shown in Fig. 2), 100 times more than the water vapor observed (Table 1). Also, injecting huge amounts of water dilutes the sulfate aerosols making them unrealistically large. The LrgH2O case increases the OH to 1.8 pptv, which is about 0.5

pptv higher than the surrounding air (Supplementary Fig. 4). Another possible way to change OH is to mimic the effect of aerosol scattering light on OH production rate. We empirically increase the OH photolysis rate by 50 times for the first 3 days (the LrgOH case in Table 1, not shown in Fig. 2) in order to fit the observed $SO_2$ trend. However, such a drastic increase of OH is unlikely to be caused only by scattering. Both the LrgH2O and the LrgOH cases are implausible explanations to shorten the lifetime of $SO_2$ for this eruption.

In addition to the $SO_2$ gas-phase reaction with OH, laboratory experiments report $SO_2$ reacting on volcanic ash and mineral dust[15–18,26,27] (Supplementary Note 3). In general, the laboratory data show a rapid initial $SO_2$ uptake, that saturates over time. We are able to estimate the uptake efficiency using the rate of decline of the $SO_2$ concentration mostly during the first day, and the saturation coverage using the time at which the $SO_2$ loss stops being controlled by the heterogeneous loss, after about one day (Supplementary Note 3). During this time about 20 % of the initial $SO_2$ has been removed from the gas phase. We find the uptake efficiency, $\gamma$, is about $3 \times 10^{-3}$, and the saturation coverage is about $3 \times 10^{16}$ molecules cm$^{-2}$ assuming a geometric surface area of spherical ash particles. This saturation coverage is equivalent to ~13% of $SO_2$ on mixed particles by mass in our simulation, for the particle size with the largest surface area, ~2 μm. These numbers will be useful to climate modelers simulating

volcanic clouds and assuming spherical particles, but because we assumed the particles are spherical they are not directly comparable to laboratory studies which often measure the Brunauer, Emmett and Teller (BET) surface area of volcanic ash samples. The surface area of real volcanic ash varies case by case and is expected to be larger than the spherical assumptions. The true $\gamma$ value should be reduced by the amount the true surface area exceeds that of an equal mass sphere. For our example of an oblate spheroid particle discussed below, the surface area is 1.4 times greater, so the $\gamma$ value would be about 2 ×

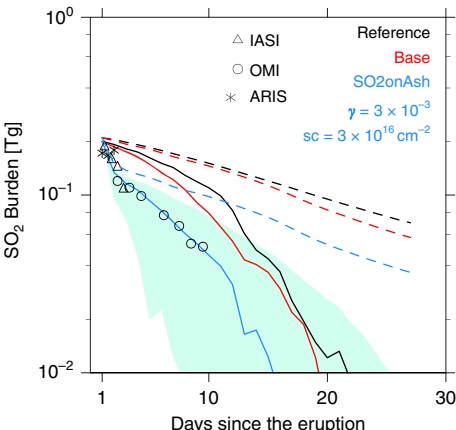

**Fig. 2 The volcanic SO2 burden.** The satellite observations are shown in symbols and the simulations are shown in lines. IASI is the Infrared Atmospheric Sounding Interferometer. ARIS is Atmospheric Infrared Sounder on board NASA's aqua satellite. OMI is Ozone Monitoring Instrument upper tropospheric and stratospheric $SO_2$ column data. The dashed lines and solid lines in the same color refer to the same simulations, but the solid lines show the simulated $SO_2$ burdens counting only the grid cells where $SO_2$ is above OMI noise level (0.2 Dobson Unit (DU), while the dashed lines show the simulated total $SO_2$ burden without accounting for observational bias due to signal to noise ratio. The boundaries of the green shaded area are two extreme estimations of simulated $SO_2$ burden of the SO2onAsh case considering the observational uncertainty. The lower boundary only counts the grid cell with the gridbox average $SO_2$ higher than the detection limit (a.k.a. five times OMI noise level). The upper boundary assumes the simulated $SO_2$ occupies only a part of the model grid cell equivalent to the five adjacent pixel threshold of OMI and compares that with the OMI detection limit. The fraction is not smaller than 5 OMI observation pixels. For details of applying observational uncertainties to the modeled results refer to Supplementary Note 1. The descriptions of the model cases are given in Table 1. $\gamma$ stands for uptake efficiency; sc stands for saturation coverage.

$10^{-3}$, but the true surface area is likely larger because of other features such as surface roughness and/or secondary phases, which can raise specific surface area by one or two orders of magnitude[18,28–30]. Usher et al.[18] reported an initial uptake efficiency in the range $7.0 \times 10^{-5}$ to $5.1 \times 10^{-4}$ for their mineral samples studied in the laboratory assuming the BET surface area of the particles. Other studies have reported lower initial uptake efficiencies in the $10^{-6}$ range[26,27]. Maters et al.[15] used a Knudsen cell to measure the initial uptake efficiency of $SO_2$ on dacite glass to be $1.3 \times 10^{-2}$ at ~250 K assuming the geometric surface area of the sample holder. Our derived values for Kelut ash assuming the geometric surface area of the particles falls between these laboratory studies. Several papers in the literature report saturation coverage of $SO_2$ on the mineral dust or ash ranging from $10^{11}$ to $9 \times 10^{14}$ molecules $cm^{-2}$ assuming the BET surface area using different laboratory technologies under different exposure times, pressures, RH, and $SO_2$ concentrations[13,15–18,31,32]. Likewise, if we scale down our saturation coverage limit by 1 or 2 orders of magnitude, the value falls between these laboratory data. Several different combinations of uptake efficiency and saturation coverage can also explain the observed $SO_2$ burden (Supplementary Fig. 5). Supplementary Fig. 5 indicates that as we increase the uptake efficiency, we can decrease the saturation coverage to fit the observation. Supplementary Fig. 5 suggests a reasonable saturation coverage, assuming spherical particles and the geometric surface area, is in the $10^{16}$ molecules $cm^{-2}$ range, and $\gamma$ is in the $10^{-3}$ to $10^{-2}$ range.

Figure 2 displays the SO2onAsh case with $\gamma = 3 \times 10^{-3}$ and saturation coverage of $3 \times 10^{16}$ molecules $cm^{-2}$. After saturation, we set the $SO_2$ uptake efficiency to zero. Figure 2 shows the modeled actual $SO_2$ lifetime, in which all $SO_2$ in the gas phase is counted and the heterogenous reaction is included, is 17 days (blue dash line). We apply the OMI observational uncertainties on the SO2onAsh case (blue solid line and green shaded area). The uncertainties increase quickly and overlap the Reference case and Base case after 10 days, which is consistent with the end time of the OMI observations when the satellite is no longer able to observe $SO_2$.

In the following sections, we continue to use results from both the Base case and the SO2onAsh case to explore the similarity and the differences of volcanic aerosol properties and ash/S interactions between cases with longer and shorter $SO_2$ lifetime.

**The persisting ash-containing volcanic aerosol layer.** CALIPSO observed the aerosol backscatter to be larger than $0.01\ km^{-1}sr^{-1}$ above Mt. Kelut two hours after the eruption. Afterward, the backscatter decayed over more than 1 month as illustrated in Fig. 3. Vernier et al.[2] used depolarization data to show that the

**Table 1 The model cases and the parameters related to SO₂ injection, as well as the SO₂ lifetime of each case.**

|  | Injection latitudes | Injection height range | H₂O injection | SO₂ lifetime (days) | SO₂ lifetime (SO₂ > 0.2 DU)[d] |
|---|---|---|---|---|---|
| Reference | 8°S | 17–26 km | no | 26 | 13 |
| Base | 12°S-2°S | 17–26 km[b] | 0.26 Tg[c] | 22 | 11 |
| SO2onAsh | 12°S-2°S | 17–26 km[b] | 0.26 Tg[c] | 17 | 7 |
| LrgH2O | 12°S-2°S | 17–26 km[b] | 26 Tg | 18 | 8 |
| LrgOH[a] | 12°S-2°S | 17–26 km[b] | 0.26 Tg[c] | 17 | 7 |

[a]The OH photolysis rate is 50 times more than the Base case in the first 3 days.
[b]In the Reference case we inject SO₂ uniformly from 17 to 26 km. In the other cases, we inject ~80% of the mass of SO₂ into 17–19 km, ~8% into 20–22 km, and ~2% into 23–26 km based on MLS SO₂ observations.
[c]H₂O injection is based on MLS H₂O observation. MLS observed about 5–8 ppm of H₂O from 85 hPa to 31 hPa above Mt. Kelut on 13 February. The volcanic H₂O injection is 2.5–4 ppm after deducting the background water vapor. MLS did not observe H₂O at lower altitudes due to obscuration by volcanic ash. In the Base case, we inject H₂O from 17 to 23 km with ~4 ppm in each model levels, with a total mass of ~0.26 Tg.
[d]Supplementary Note 1.

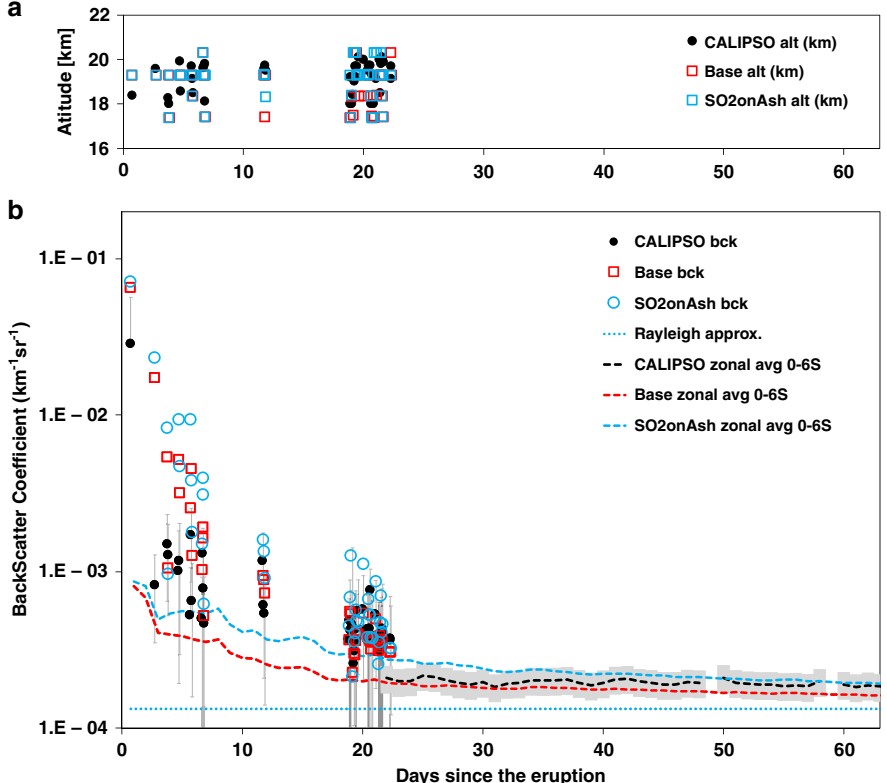

**Fig. 3 the volcanic ash layer altitudes and the total backscatter coefficient. a** the volcanic ash layer altitudes from CALIPSO and two model cases. **b** The total backscatter coefficient (aerosol + Rayleigh). We pick the largest total backscatter in the vertical profiles and its corresponding altitude on the selected segments to define the altitudes in panel a and the values in (**b**) (dots and circles). The dashed lines are the zonal average backscatter from CALIPSO (black) and two model cases (red and blue). CALIPSO does not have observations from 21 Feb to 23 Feb and from 25 Feb to 2 March. The blue dotted line represents the approximate background backscatter from air plus the background sulfate aerosol. The gray error bars represent the standard deviation of the dotted CALIPSO data. The gray shaded area shows the modeled average values are within 20% of CALIPSO averaged values. The standard deviation of the CALIPSO zonal average values is over a factor of 200%.

non-spherical particle signal is close to zero 110 days after the eruption. Vertically, the volcanic aerosol had a three-layer pattern the first day after the eruption. After that, CALIPSO only detected one obvious layer near 18–21 km. Generally, the altitude of the volcanic aerosol layer to the east of Kelut was lower (17–19.5 km) and the ash layer to the west was higher (18–20 km), indicating wind shear caused the volcanic aerosol layers at different altitudes to move in different directions.

In order to match the simulated backscatter coefficient values and the vertical distributions with CALIPSO observations, we use the method described by Vernier et al.[2] to choose the orbit segments in which CALIPSO observed the volcanic aerosol layers on each day. For each segment, we average the total backscatter (Rayleigh + all types of aerosols) to create a vertical profile. In Fig. 3, we pick the altitudes and the values with the largest total backscatter in the vertical profiles from CALIPSO (black) and the simulations (red and blue) from 13 Feb to 4 March. The altitudes of the simulated ash layers are consistent with the CALIPSO observed altitudes when we use an initial injection height from 72 to 52 hPa (18.5–20.5 km). The altitude differences between the simulations and observations in Fig. 3 can be caused by the mismatch of the vertical resolution of the model, which is about 1 km in the stratosphere, and the CALIPSO data, which is about 60 m. The backscatter data from CALIPSO (black) decreases rapidly in the first few days and then decreases more slowly. The simulated values decrease more slowly from day 1 to day 5 than the observations. However, the Base and the SO2onAsh cases are generally within the standard deviation of the observed values

after day 6. We conducted several tests with various initial size distributions and particle number densities, but they all show similar decreases in backscatter with time.

It is tricky to compare simulations with the CALIPSO data in the first few days. CALIPSO may have missed volcanic aerosol layers in the first few days because the orbit may not have crossed narrow layers. The mesoscale winds are also important for aerosol spreading in the first couple of days, but are not well represented in the model due to the coarse horizontal resolution as we discuss in the Supplementary Note 2 and Supplementary Fig. 1.

After 20 days, the volcanic aerosols are generally well spread in longitude. Figure 3 also shows the zonal average backscatter from CALIPSO (black dash line) and compares it with the simulated zonal average backscatter (red and blue dash line). These three lines correspond to an of altitude 19.5 km and a latitude range of 0–6° S, where both the observations and the simulations have the largest zonal average backscatter signals 20 days after the eruption. After 40 days, the differences between the simulated values (red and blue dash lines) and the observation (black dash line) are <20% and all of them are slightly higher than the background (background aerosol+rayleigh) backscatter, indicating the presence of the volcanic aerosols.

**Particle size distribution.** Instruments onboard the NASA Global Hawk measured the size distributions of volcanic ash near Guam above 16.5 km altitude during ATTREX flights from 6 March to 12 March, 20–25 days after the eruption[3]. Measurement

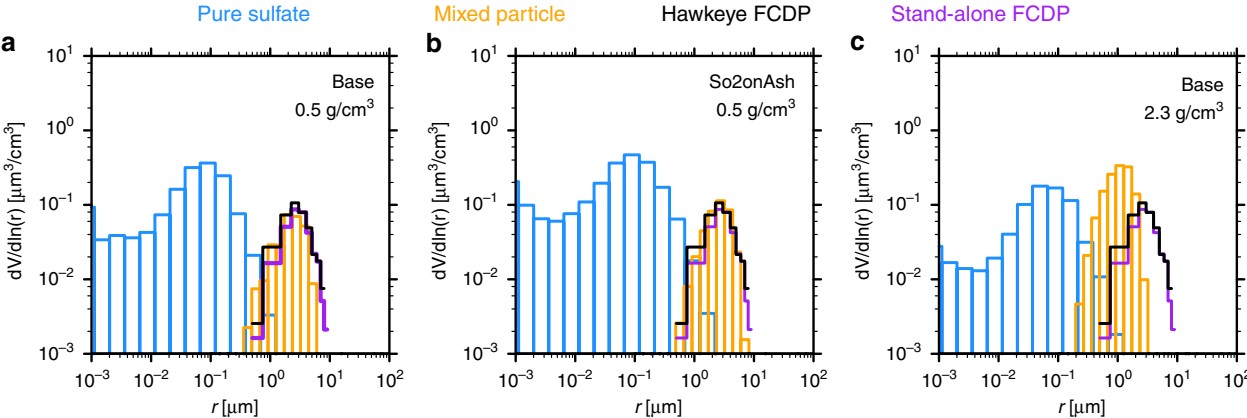

**Fig. 4 The particle volume size distributions near Guam.** The particle volume size distributions from the simulated pure sulfate (blue), mixed particles (orange) and the two ATTREX FCDP probe observations (black and purple) around 16.5–18.5 km near Guam about 25 days after the eruption. No pure ash remains at this time. Panel a is the Base case assuming ash density of 0.5 g/cm³; **b** the SO2onAsh case assuming ash density of 0.5 g/cm³; **c** the Base case assuming ash density of 2.3 g/cm³.

details are described in Woods et al.[33] and Jensen et al.[3]. These observations constrain the input ash size distribution and the particle density or shape. As noted by Durant et al.[8], volcanic particles are not spherical and particle densities are determined by the particle compositions. Densities of glassy volcanic particles are around 2.3 g cm⁻³, while those for pumice are about 0.6 g cm⁻³. The ash particle density and shape are important to the simulations of sub- and super-micron-sized mixed particles because they affect the fall velocity, which is large enough for the super-micron-sized particles to experience significant sedimentation in 20 days. A spherical particle with a radius of 5 μm with a particle density of 0.6 g cm⁻³ falls at about the same speed as a particle of 2.3-μm radius with a particle density of 2.3 g cm⁻³. Highly oblate shapes have effects on fall speeds comparable to spherical shapes with similarly lower densities. Our model tests (Supplementary Fig. 6) indicate particles with length/diameter ratios of 0.3 only reduce the fall velocity by 8% compared with the equivalent-volume spherical particle, while highly oblate particles with length/diameter ratio of 0.1 can reduce the fall velocity by 30%.

We ran simulations assuming various mass burdens in three ash size modes and/or with various ash particle densities to find the best choices to match the ATTREX observations (detailed in "Methods"). Figure 4 shows the simulations of the volume size distributions of pure sulfate (blue) and mixed particles (orange) in the Base and SO2onAsh cases compared with ATTREX data from two FCDP probes (black and purple). Modeled ash particles are spherical with various densities of 0.5 and 2.3 g cm⁻³ (Fig. 4). The simulation results in Fig. 4 (blue and orange) are sampled near 170° E and 8° S on 10 March from 16.5 to 18.5 km. This location matches the ATTREX flight path on 9 March and 10 March. We find the particle density is the main controller of the size distribution on 10 Mar (25 days after the eruption). Decreasing the particle density from 2.3 g cm⁻³ (Fig. 4c) to 0.5 g cm⁻³ (Fig. 4a) significantly improves the observed (black and purple) and simulated (orange) particle size distribution agreement around 5 μm, and also limits the particle numbers near 0.8 μm as shown in Supplementary Fig. 7. The size distribution of the base case and the SO2onAsh case are very similar. Note that the actual mixed particle densities are slightly different from the ash particle densities because of the mixing with sulfate. The mixed particle densities for these three panels in Fig. 4 are 0.52, 0.57, and 2.25 g cm⁻³. The balance between mode 1 (~0.5 μm) and mode 2 (~1.5 μm) in the initial size distribution (see Methods) is also relatively important. If we put the majority of mass in mode 1, the

particles do not have enough time to grow/coagulate and reproduce the size distribution above 5 μm on 10 March. We also show the number size distribution for each case in Supplementary Fig. 7, which shows an agreement with the observation assuming an ash density of 0.5 g cm⁻³. Note that, in reality, ash particles are usually non-spherical with a non-porous structure or with a porosity dominated by macropores (diameters > 50 nm)[29]. Our conclusion is that the non-spherical, porous or aggregate shaped particles have the same fall velocities as spherical particles with densities of 0.5 g cm⁻³.

Figure 5 shows the evolution of the normalized size distribution of the mixed particles as a function of time. The Base case and the SO2onAsh case show the volume density of mixed particles peaks near 2–5 μm in radius after day 11 and the large particles (above 5 μm in radius) fall out through the time period. Comparing the Base case and the Nocoag case, we show that the sub-micron particles coagulate into larger radii particles especially in the first ten days. Coagulation of mixed particles with ash and pure sulfates is the key process needed to produce the evolution of the size distribution of the mixed particles. After 11 days, the shape of the size distribution does not change very much. Supplementary Fig. 9 shows the pure sulfate size distribution evolution. Condensational growth and coagulation increase the pure sulfate size from 0.1 to 0.3 μm in a month.

**Impact of volcanic ash on sulfate burden.** Volcanic ash can accumulate S species by uptaking H₂SO₄ gas, sulfate aerosols, and SO₂ gas. Large volcanic ash particles fall out of the stratosphere quickly, which reduces the stratospheric sulfate burden. Here, we investigate the stratospheric S/ash interactions in three simulations (Fig. 6) from 33° S to 16° N: the Noashemission case (blue), the Base case (black), and the SO2onAsh case (red). They use the same SO₂ injection. First, neither the Noashemission case nor the Base case, with a S residence time of 22 days, consider SO₂ reactions on ash. The difference of S burden (Fig. 6a) between them is minor indicating the sulfate and H₂SO₄ gas is not effectively removed directly by ash since the SO₂ lifetime is long compared with the super-micron ash removal time. As we include the SO₂ reacting on ash surfaces, the sulfate produced by reactions on the ash stays with the ash particle and is removed effectively. After two months, the stratospheric S burden is reduced to 67% of its original value in the case with the heterogeneous reaction compared to 77% with the Noashemission

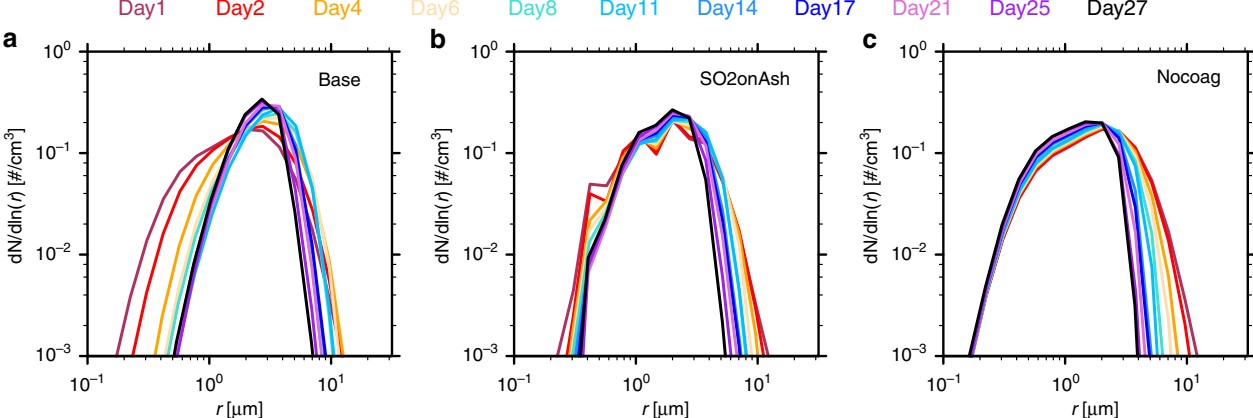

**Fig. 5 The normalized mixed particle volume size distribution at 61 hPa.** The normalized mixed particle volume size distribution at 61 hPa (19.5 km) for the Base case, the SO2onAsh case, and the Nocoag case. The Nocoag case shows the simulation without coagulation between mixed particles and pure sulfates or between mixed particles and ash. The values are picked at the location with the largest mass mixing ratio of mixed particles on the days specified on top of the figure. The un-normalized distributions are shown in Supplementary Fig. 8.

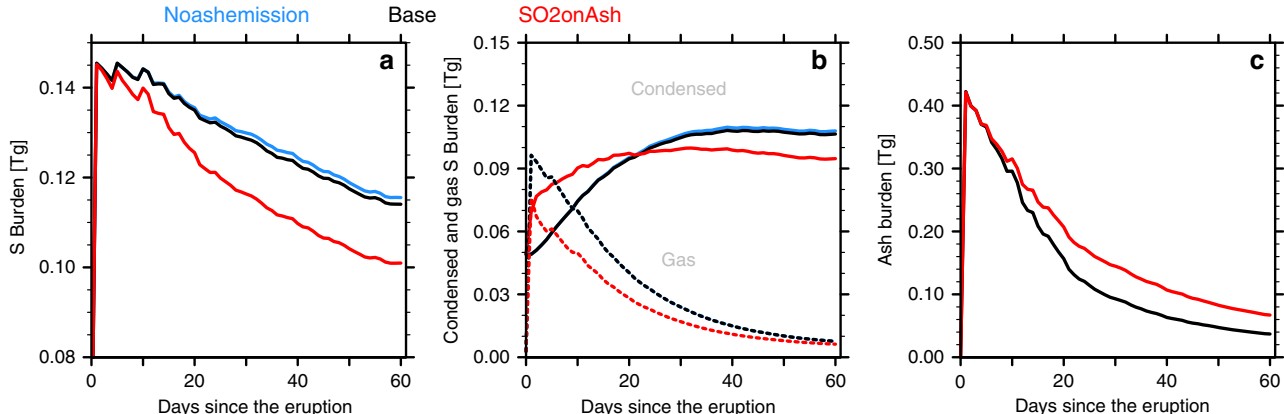

**Fig. 6 The stratospheric S burden and ash burden between 33˚S to 16˚N. a** The stratospheric S burden from the Noashemission case (blue), the Base case (black), and the SO2onAsh case (red) between 33˚S to 16˚N; **b** the condensed phase (solid) and the gas phase (dotted) S burden. The gas S burden of the Noashemission case and the base case are nearly identical; **c** The stratospheric ash burden. The stratospheric background S burden is ~0.052 Tg within this area.

case and the Base case. The amount of S removed is 43% larger with heterogeneous chemistry than without. Figure 6b shows that $SO_2$ reaction on ash (red) produces sulfate faster than $SO_2$ oxidation to sulfate in the gas phase (black and blue). The $SO_2$ reacting on ash surfaces also affects the ash burden evolution (Fig. 6c) but the impact is a change of <8% of the ash mass in 2 months. The difference can be caused by the slightly different densities and effective radii of the mixed particles between the two model cases.

Figure 7 shows the contribution of ash and sulfate to the backscatter coefficient and mass mixing ratio in the SO2onAsh case from 72 hPa to 52 hPa (19.5–21.5 km). The Mixed particle is the main contributor to the backscatter (Fig. 7b). Inside the mixed particle, the ash is the major component (6c). Ash contributes 60% of the mixed particle mass 60 days after the eruption (6e). The 40 % of the mixed particle that is sulfate has two origins: about 20% of sulfate inside the particle is from $SO_2$ reacting on ash surface and the other 20% is from coagulation with pure sulfate and $H_2SO_4$ gas growth on ash surface (6e). The majority of sulfate is in the form of pure sulfate rather than the mixed sulfate after 30 days of the eruption (6f). This dominance of pure sulfate indicates that $SO_2$ oxidation to $H_2SO_4$ in the gas phase followed

by nucleation into pure sulfate is the dominant sulfate production processes at this time since the heterogeneous reaction of $SO_2$ on ash has shut down. We also see similar contribution patterns of ash and sulfate to the total backscatter and mixing ratio in the Base case as shown in Supplementary Fig. 10. Note that the sulfate percentage in the mixed particle increases much slower in Supplementary Fig. 10e than in Fig. 7e. The slower increase explains why the ash cannot reduce the S burden effectively in the base case. Therefore, we conclude that the ash particles are the main component of the persisting volcanic aerosol layer following this eruption.

## Discussion
Our investigation of volcanic ash using the Whole Atmosphere Community Climate Model provides new insight into the role of volcanic ash in climate change. First, our simulations, combined with CALIPSO and ATTREX data, indicate that the persisting volcanic aerosol layer after the 2014 Mt. Kelut eruption is primarily composed of ash (Fig. 7 and Supplementary Fig. 10), which is likely low density as well as being non-spherical and super-micron sized (Fig. 4 and Supplementary Fig. 7). Our

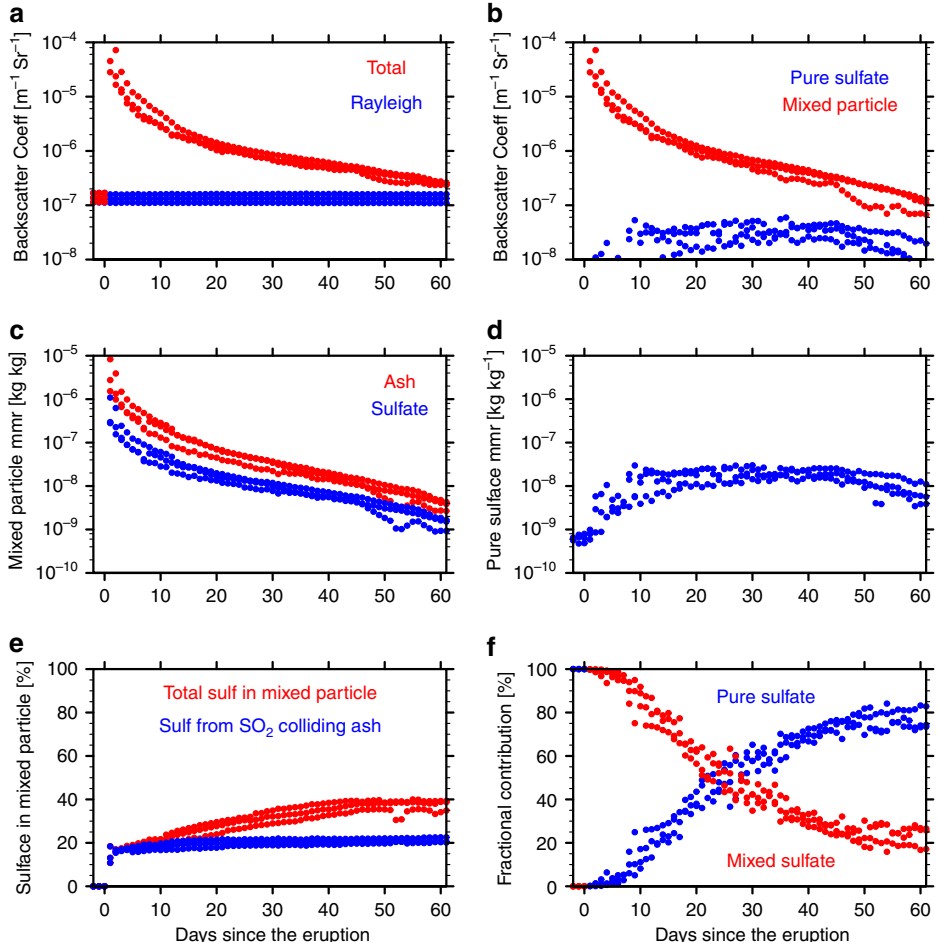

**Fig. 7 The composition evolution from the SO2onAsh case.** The data are picked from the maximum total backscatter at three different pressure levels from 72 to 52 hPa. **a** the maximum total backscatter (aerosol + Rayleigh) in red and Rayleigh backscatter in blue; **b** the backscatter contribution from the mixed particle and pure sulfate respectively; **c** the mass mixing ratio of ash and sulfate (H2SO4 + water) inside the mixed particles; **d** the mass mixing ratio of pure sulfate; **e** the weight percent of sulfate inside the mixed particle. The blue dots are the sulfate coming from SO2 reacting on ash and the red dots are the total sulfate percentage (blue dots + sulfate from coagulation/growth); **f** the weight percent of pure sulfate in total sulfate (blue) and the weight percent of sulfate in mixed particle in total sulfate (red). The Base case compositions are shown in Supplementary Fig. 10.

estimate of ash density from the simulations is about $0.5\,\mathrm{g\,cm^{-3}}$, which corresponds to pumice[8]. This low density is much smaller than assumed in most model simulations. However, the particle shape may also impact the fall velocity. We conclude that density and shape combine to produce a fall velocity equivalent to spherical particles with a density of $0.5\,\mathrm{g\,cm^{-3}}$, but we cannot determine the relative importance of shape and density. The typical ash/sulfate mixed particle size is over 10 times larger than the size of the typical stratospheric background sulfate aerosol (~0.1 μm). Observations from several large volcanic eruptions also show persisting volcanic debris[4–6], with volcanic ash sizes similar to that of the Mt. Kelut eruption. Simulations of large eruptions that include the small size mode of ash discussed in the Methods are needed to determine the contribution of ash to observed cooling at the surface. Sulfate coated ash may have a larger surface area than sulfate alone, which may influence stratospheric heterogeneous ozone chemistry[34].

We find that volcanic S can be removed from the gas phase by volcanic ash more rapidly than considered in climate models that have ignored heterogeneous reactions of $SO_2$ on ash. We find that neglecting the uptake of $SO_2$ on ash minimizes the calculated effect of ash on removing volcanic S by sedimentation in a small volcanic eruption. However, $SO_2$ reacts on ash,

as shown by laboratory experiments[15–18]. Simulations considering these reactions give a shorter $SO_2$ lifetime, agreeing better with observations following the Mt. Kelut eruption. We distribute the initial eruption plume over an extended area in order to capture the observed wind shear and plume transport. However, we find that more confined injections do not impact our conclusion that heterogeneous reactions of $SO_2$ on ash are important. We also find that some of the short lifetimes reported following smaller eruptions[20] may be caused by satellites missing some $SO_2$ due to signal to noise issues as the debris layer spreads out. The $SO_2$/ash reactions enhance the S/ash interactions and favor more sulfate removal by ash. While our simulations are focused on a relatively small eruption, Guo et al[35,36] reported that half of the $SO_2$ released to the stratosphere in the Pinatubo eruption was converted to sulfate within 3 days of the eruption.

Further studies of the heterogenous chemical reaction of $SO_2$ on ash following other large and small eruptions are warranted since the ratio of ash to $SO_2$ injections is variable. Also, climate models with higher resolution can reproduce the spread of volcanic ash and $SO_2$ better. Finally, climate models need to better quantify the contribution of volcanic ash particles to the energy balance and heterogeneous ozone loss.

## Methods

**Model setup.** We use the specified dynamics (SD) version of the Whole Atmosphere Community Climate Model (WACCM) version 4.0[37,38] coupled with the Community Aerosol and Radiation Model for Atmospheres (CARMA)[39–45]. The CARMA model has been used to treat many different types of stratospheric aerosols in global model frameworks: sulfate aerosol[42], volcanic sulfate aerosol[45,46], polar stratospheric clouds[43,44] as well as noctilucent clouds and micrometeorites[39,41]. The model horizontal resolution is 1.9° latitude × 2.5° longitude. The vertical resolution is about 1 km in the lower stratosphere.

For this study, we further develop the sulfate aerosol and micrometeorite model employed by English et al.[42] and Bardeen et al.[39] to be able to simulate volcanic ash particles and their interactions with pure particles. The model considers 22 particle mass bins for each type of particle covering particle radii from ~0.088 to ~63 μm for ash particles and for mixed particles, and from ~0.34 to ~72 μm for pure sulfate aerosol. In total there are 66 particle bins being advected. The sulfate density depends on the water fraction of the particle. The ash density varies from 0.5 g cm$^{-3}$ to 2.3 g cm$^{-3}$ for different test cases. The mixed particles contain non-volatile ash as well as $H_2SO_4$[47], along with liquid water that is assumed to be in equilibrium with atmospheric humidity. In the model, the particle density is variable, so the actual radius values covered are variable. The main microphysical processes (Fig. 1) include heterogeneous nucleation of $H_2SO_4$ gas on ash particles; homogeneous nucleation of $H_2SO_4$ gas[48]; coagulation of sulfate particles, ash particles and mixed particles; the growth and evaporation of $H_2SO_4$ and $H_2O$ on/from sulfate aerosol and mixed particles; particle sedimentation and dry deposition. The WACCM model deals with S chemistry[21,42], particle transport by winds and wet scavenging. Figure 1 also shows the main $SO_2$ chemistry we are considering in this paper: $SO_2$ uptake on ash and $SO_2$ reaction with OH. $SO_2$ aqueous phase chemistry is also important for tropospheric volcanic $SO_2$ oxidation but is not included here since liquid water is not expected in the stratosphere[14,49], and since we focus on $SO_2$ and stratospheric volcanic cloud evolution days and months after the eruption. However, including the effects of these tropospheric reactions on the volcanic plume and cloud would be valuable to better understand the total sulfate deposition on ash, some of which occurs before the ash enters the stratosphere[9–13].

**Observational data.** The Cloud-Aerosol Lidar with Orthogonal Polarization (CALIOP) instrument on board the CALIPSO satellite measures the backscatter at 532 and 1064 nm[50]. We use the CALIPSO L1B nighttime total backscatter coefficient data at 532 nm. In the stratosphere, the vertical resolution can be as high as 60 m from 15 to 20 km and 180 m from 20 km above. For nighttime observations, the uncertainty due to noise is estimated to be typically smaller than 1%.

The ATTREX field campaign applied the Fast Cloud Droplet Probe (FCDP)[33,51,52] to measure the forward scattered light from individual particles. The instrument retrieves particle sizes ranging from 1 to 50 μm diameter. We use the size distribution data published by Jensen et al.[3] from two onboard FCDP probes: the

stand-alone FCDP and the Hawkeye probe. The particles were mainly detected during flights between 6 March and 12 March and between 16.5 and 18.5 km.

The OMI reported volcanic $SO_2$ column data starting 14 February, one day after the Mt. Kelut eruption. We use the OMI L2 data that only counts the upper tropospheric and stratospheric SO2 columns. The background noise level of this dataset is 0.2 DU.

The MLS onboard NASA's Earth Observing System (EOS) Aura satellite observes $H_2O$ concentrations. We use the MLS v4.2 $H_2O$ data as a reference for volcanic water injection on Feb. 13th, 2014. We refer to the water mixing ratio from 82 hPa to 31 hPa (~17 km to 23 km) near Mt. Kelut.

**The volcanic emission references.** Volcanic eruptions inject various particles and gases into the atmosphere, such as ash, $H_2O$, $CO_2$, $SO_2$, HCl, $H_2S$, etc. For this study, we constrain the $SO_2$ and volcanic ash emissions. Several satellite observations provide estimates of $SO_2$ injection on 13 February varying from 0.12 Tg to 0.8 Tg available at Global Sulfur Dioxide Monitoring Home Page. However, the ash masked the $SO_2$ and made it difficult to determine the injection of $SO_2$. Neely and Schmidt[53] compiled a database of climatically relevant volcanic $SO_2$ emissions since 1990. It suggests an injection of 0.3 Tg of $SO_2$ uniformly mixed from ~17 to 26 km on 13 Feb, 2014 at 7.93°S and 7.93°E. We use this database as a reference for the injection height and inject over 6 h, following Mills et al.[21]. But we find the best estimate of the $SO_2$ injection is 0.2 Tg based on comparing simulations with the OMI data.

Kristiansen et al.[54] constrained the ash emission for particle radii from 1.4 to 14 μm with a total injection of 0.38 Tg into the stratosphere based on the ash retrieved from Japan's second Multifunctional Transport Satellite (MTSAT 2) on Feb 13 and 14. We apply a different approach to constrain the ash emission for the global modeling framework which focuses more on the long-lived sub- and super-micron-sized volcanic ash. We use a 3-mode set of lognormal size distributions for our initial ash emission estimation (Fig. 8). The first mode has a mean radius of 0.476 μm and a variance of 0.535 based on aircraft size distributions measured in the Mt. St. Helens volcanic cloud of 1980[7]. The second mode has a mean radius of 1.5 μm and a variance of 0.5 fitting the Kelut retrieval of Kristiansen et al.[54]. The third mode is centered at 20 μm with a variance of 0.8. This mode is suggested from ash deposits near Mt. St. Helens[8]. Even larger ash particles were likely injected[8], but these would have such short lifetimes that they are neglected here.

We vary the total mass of ash, the percentage of the particle mass in each of the three size modes, the ash particle density, the altitude range of the injection, and the geographic location of the injection to best agree with CALIPSO satellite and ATTREX aircraft data. The best estimation of ash emission in the Base case is detailed in Table 2.

We calculate the backscatter at 532 nm of gas (Rayleigh), pure sulfate aerosol using a T-matrix code[55] and spherical ash and mixed particles[56]. The refractive indexes are assumed to be 1.32 for pure sulfate[57], and 1.55 + 0.001i for mixed particles and ash[58].

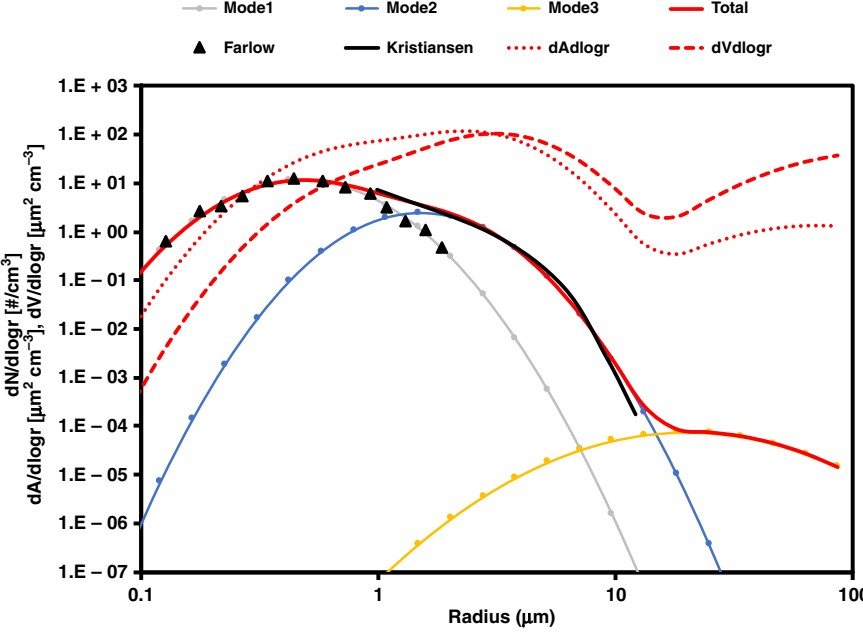

**Fig. 8 The initial size distribution of ash.** The gray line represents mode 1 and fits the black triangles by Farlow et al.[7]. The blue line represents mode 2 and fits the black line by Kristiansen et al.[54]. The yellow line represents mode 3 based on Durant et al.[8]. The red solid line is the sum of the three modes and is the number size distribution used for initial injection of ash. We also show a red dotted line and a red dash line representing the area size distribution and volume size distribution for the initial ash injection.

**Table 2 The Ash parameters in the Base case and the SO2onAsh case.**

| Ash injection parameters | Descriptions and values |
|---|---|
| Injection amount | 0.45 Tg (0.058 Tg in mode 1, 0.305 Tg in mode 2, and 0.087 Tg in mode 3) |
| Ash density | 0.5–2.3 g cm$^{-3}$ range of variation in tests |
| Ash shape | Spherical |
| Size distribution | Mode 1: rm = 0.476 μm, and a variance of 0.535 |
| | Mode 2: rm = 1.5 μm, and a variance of 0.5 |
| | Mode 3: rm = 20 μm, with a variance of 0.8 |
| # of latitude grid boxes | 5 grid boxes (from 2.8˚ S–10˚ S) used for injection |
| Vertical distribution | 3, 1 km thick layers centered at 18.5 to 20.5 km with 68% at 19.5 km and 16% at both 18.5 and 20.5 km. |

Volcanic ash is often neglected in climate simulations as it is assumed to have a short atmospheric lifetime. Here, the authors show a persistent super-micron ash layer after the Mt. Kelut eruption in 2014 that impacts the stratospheric sulfur burden and chemistry for over the first months after the eruption.

We interpolate the simulated results onto CALIPSO orbits. We find the best ash emission size distribution for the injection as shown in Fig. 8 and Table 2.

## Code availability

The CESM/CARMA model is available on the CESM trunk to any registered user at www.cesm.ucar.edu.

## Data availability

The satellite data in Fig. 2 are from Global Sulfur Dioxide Monitoring Home Page [https://so2.gsfc.nasa.gov/]. The CALIPSO data and the OMI data are available at https://search.earthdata.nasa.gov. The ATTREX data are available at [https://espoarchive.nasa.gov/]. The Integrated Global Radiosonde Archive data are available at [https://www.ncdc.noaa.gov/]. The MLS data are available at [https://mls.jpl.nasa.gov/]. The main data generated during this study are available at [https://osf.io/8cpd3/] with a permanent https://doi.org/10.17605/OSF.IO/8CPD3. Additional information is available from the corresponding author on reasonable request.

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

## Acknowledgements
The work at the University of Colorado was supported by NASA grant NNX16AQ37G, NASA grant 80NSSC19K1276, and National Science Foundation (NSF) grant PLR1643701. We thank Dr. Mingxuan Wu from the University of Wyoming for providing the CALIPSO orbit data to help to convert model output in the CALIPSO orbit format. We thank Dr. Ping Yang, Dr. Bingqi Yi, and Jiachen Ding for their help with the particle optical properties. We appreciate discussions with Darya Urupina, Dr. Russ Schnell, graduate student Margot Clyne, Professor Anja Schmidt, and Dr. Anna Kristiansen. WACCM is a component of NCAR's Community Earth System Model, CESM, which is supported by NSF and the Office of Science of the U.S. Department of Energy. We acknowledge high-performance computing support from Cheyenne (doi:10.5065/D6RX99HX) provided by NCAR's Computational and Information Systems Laboratory (CISL), sponsored by the NSF. Data storage supported by the University of Colorado Boulder "PetaLibrary".

## Author contributions
O.B.T., E.J., and Y.Z. discussed the hypothesis. Y.Z., O.B.T., and M.T. wrote the manuscript. M.T. summarized the laboratory study. E.J. S.W. provided the ATTREX data. Y.Z., P.Y., C.B., and M.M. participated in the model development. Y.Z. performed the model simulation and analyzed the observational data. All the authors edited the manuscript.

## Competing interests
The authors declare no competing interests.
