## [Peer Review File · Nature Communications]

Reviewers' Comments:

Reviewer #1:

Remarks to the Author:

SUMMARY

This modelling study addresses a very interesting question regarding the impact of volcanic ash on the stratospheric S burden, using the 2014 Mt. Kelut eruption as a case study. Comparing the output of various model simulations with satellite observations following this eruption, the authors conclude that including SO₂ reaction on ash particles produces the best match to real observations of stratospheric SO₂ lifetime. This is in addition to the interaction of ash particles with SO₄²⁻ and H₂SO₄ before fallout. Further, they suggest that a very low ash density of 0.5 g/cm³ can explain the ash persistence in the stratosphere, based on comparison to aircraft particle size distribution observations. The study represents a novel assessment of the importance of volcanic S/ash interactions in the stratosphere, and could ultimately represent a high impact publication contributing to new knowledge in this field.

However, as it currently stands, it seems that incorrect values of SO₂ uptake on ash are used in the model simulations, several key SO₂ oxidation pathways in the atmosphere are not included in the model or even acknowledged, the various S species and their interaction with ash are not always expressed in a clear or consistent manner, and a difference in ash shape (from spherical) is not considered as a factor potentially contributing to ash persistence in the stratosphere.

Overall these issues, in addition to numerous minor points, lead me to recommend rejection of this manuscript. However, if the authors are able to confirm their findings using the correct SO₂ uptake values, I would encourage them to consider resubmission.

MAJOR COMMENTS

- The uptake efficiency and saturation coverage values used in the model simulations (lines 144-147), based on the laboratory data of Maters et al. (2017), are off by 2 orders of magnitude in opposing directions. The authors correctly assume that only 'the geometric surface is available for initial uptake' (line 146). However, they then use an initial uptake efficiency value on the order of 10⁻⁴ normalized to the BET surface area, when in fact they should use a value on the order of 10⁻² normalized to the geometric surface area (as reported in lines 90-92). The authors then incorrectly 'assume the geometric surface area' in considering the saturation coverage (line 147), when in fact SO₂ should be able to access both external and internal (BET) surfaces over the saturation timescale. Hence, they use a saturation coverage value on the order of 10¹⁵ cm⁻² normalized to the geometric surface area, when in fact they should use a value on the order of 10¹³ cm⁻² normalized to the BET surface area (as reported in lines 94-95 and 148-150). Thus, the claim (line 151) that the 'assumptions are similar to those used in Maters et al. for volcanic ash' is incorrect, and unfortunately an initial uptake efficiency 2 orders of magnitude too low and a saturation coverage 2 orders of magnitude too high are used in the model simulations. The correct values will need to be adopted to check if the revised model simulation results allow the same conclusions to be drawn regarding the importance of SO₂ reaction on ash in this study.

- Excluding the heterogeneous reaction of SO₂ on ash, only the gas-phase reaction of SO₂ with OH is acknowledged and modeled here as an SO₂ oxidation pathway in the volcanic plume and/or in the atmosphere (lines 70-73, 82, 122-133, Table S1, Figure 1), whereas the liquid-phase reactions of SO₂ with H₂O₂, O₃, and O₂ (catalysed by transition metal ions) are ignored despite also being key SO₂ oxidation pathways. These latter reactions might be as important as SO₂ reaction with OH following an eruption, especially as the water-rich plume might provide significant amounts of water locally to the stratosphere, and this should be included in the model simulations here as well as acknowledged with reference to highly relevant recent literature on this topic (e.g., Martin, 2018; Galeazzo et al., 2018).

- There seems to be confusion in the way that various S-bearing species as well as their interaction with volcanic ash are expressed throughout the manuscript, which unfortunately translates to the reader a lack of clarity of the processes being discussed. The authors could better distinguish the different S chemistry pathways and the role of ash in this context. For example, various terminology is used to describe SO₄²⁻/H₂SO₄ interaction with ash including 'growth on', 'coagulation on', and 'nucleation on' (e.g., lines 303-304, 381-383), which refer to different processes but this is not well explained. Including Figure S9 in the manuscript instead of the supplement could partly help here. Moreover, the uptake of SO₄²⁻/H₂SO₄ on ash surfaces (by whatever process) is distinct from the formation of SO₄²⁻/H₂SO₄ on ash surfaces following the uptake and oxidation of SO₂ gas. Clarifying this distinction is important because it means that, even if volcanic SO₂ gas has not yet been oxidized to SO₄²⁻/H₂SO₄ in the atmosphere (by gas- or liquid-phase reactions), ash is still able to act as a sink for volcanic S by reacting with the SO₂ gas. This comes across in some parts of the text but not others. For instance, why is SO₂ gas and ash reaction not included in Figure S9?

Additionally, there is inconsistency throughout the text in the use of the chemical compound names versus their chemical formulas. For example, 'sulfur dioxide' is never defined and only 'SO₂' is used, 'sulfuric acid' and 'H₂SO₄' are interchanged throughout, and 'sulfate' is used and never defined as 'SO₄²⁻'.

Relatedly:

Lines 67-69: This sentence is confusing as written. First, 'the longer SO₂ takes to convert to H₂SO₄' is probably more accurate as 'the longer it takes SO₂ to be converted to H₂SO₄'. Second, it is not entirely clear how 'the less H₂SO₄ vapor is available to grow onto ash' and 'the fewer sulfuric acid particles are available to coagulate on ash particles' differ, please explain the distinction between these two modes of H₂SO₄ uptake on ash.

In addition, the entire statement seems to neglect that even if SO₂ gas has not yet been converted to SO₄²⁻/H₂SO₄ (by gas- and liquid-phase oxidation pathways), ash particles could still play a role in capturing volcanic S by reacting directly with SO₂ gas before falling out of the stratosphere.

Lines 277-281: What is meant by 'growth of sulfuric acid gas on ash'? Ash can also take up volcanic S by reaction with SO₂ gas but this is not mentioned in the opening line. However, later the 'SO₂onAsh case' is then included in what the authors refer to as 'sulfate/ash interactions' investigated in model simulations, but as written this does not encompass SO₂/ash interaction. The broader 'sulfur/ash interactions' could be more accurate.

Lines 283: 'sulfate cannot be effectively removed by ash when the SO₂ lifetime is long' - to avoid any misunderstanding please state explicitly that this is when reaction between SO₂ and ash is neglected. Also, is it just 'sulfate' or also 'sulfuric acid'?

- It appears that only variation in ash density is considered to explain the Mt. Kelut ash persistence in the stratosphere (lines 50-53, 138-140, 325-327, Table S2), whereas ash shape could also be a contributing factor and variation in this physical property should similarly be investigated in the model simulations and compared with the ATTREX observations. In other words, alongside of challenging the assumption that ash has a density of ~2.3 g/cm³ (as done in this study), it would be as valid to challenge the assumption that ash is a spherical particle (line 239, Table S2). Without doing this, it seems that various possibilities to explain the airborne ash lifetime have not been fully explored here, and the conclusion that a low ash density must be responsible is seriously weakened. Relatedly:

Lines 55-56: Model simulations are performed which show that an ash density of 0.5 g/cm³ delivers the best match to particle size distributions observed (ATTREX flight measurements) after

the Mt. Kelut eruption. It is not entirely clear if variation in only this one physical property (ash density) is tested to explain the persisting ash in the stratosphere. Did the authors consider other physical properties such as ash shape (despite lines 232-233 mentioning that a shape change from spherical to oblate could reduce ash fall speed)? Lines 429-431 state "we vary the total mass of ash, the percentage of the particle mass in each of the three size modes, the ash particle density, the latitude range of the injection and the geographic location of the injection to best agree with CALIPSO satellite and ATTREX aircraft data." Maybe this is what the authors refer to when claiming to investigate different physical processes?

- Lines 19-20: Is the claim that ash removes 50% more S correct? Lines 286-288 state that 'In two months, the stratospheric S burden is reduced to 51% of its original value in the case with the heterogeneous reaction [on ash] compared to 68% with the Noashemission case and the Base case.' To me this implies that the addition of S chemistry on ash only removes 17% (not 50%) more volcanic S from the stratosphere compared to if these heterogeneous reactions are ignored.

MINOR COMMENTS

Line 1: The title could be improved. First, the term 'stratospheric extinction' does not appear anywhere in the manuscript, and its meaning here is unclear. Second, this study does not actually provide new knowledge on the role of ash in 'SO₂ chemistry', but incorporates existing data on ash-SO₂ reaction in model simulations to reveal an impact of ash on 'SO₂ lifetime' in the stratosphere.

Line 25-27: It seems misleading to include 'volcanic minerals, referred to as ash' here in describing the volcanic aerosols that 'have altered climate throughout Earth's history'. Traditionally, the stratospheric volcanic aerosols considered to affect climate (by processes mentioned in lines 27-29) refer to liquid sulfuric acid/sulfate aerosols, not to ash particles. Further, the term 'volcanic mineral' is inaccurate since volcanic glass (which is non-crystalline) is typically a major component of ash.

Line 31: 'large, high-density' is vague - please clarify/define.

Line 34: 'relatively small' is vague - please clarify/define (e.g., using the Volcanic Explosivity Index).

Line 43: 'volcanic ash was observed' - please insert 'in the stratosphere' to clarify the context.

Lines 45-46: Please provide a reference related to the 'Mt. St. Helens in 1980' case. Also, please clarify what specifically is meant by 'available ash observations' (e.g., 'by satellite'), since ash is often 'observed' in general during or after an explosive eruption.

Line 50: Please provide a reference for the value of 2.3 g/cm³. Typically a value closer to 2.5 or 2.6 g/cm³ is reported, and 2.5 g/cm³ is also quoted in this study on line 140.

Line 59: Please insert 'that' between 'show' and 'there is'.

Line 60: 'a process previously overlooked' - please clarify the context you refer to (e.g., 'in models'). The notion that airborne volcanic ash/glass can capture SO₂/SO₄²⁻/H₂SO₄ is well recognized in the literature (e.g., Rose, 1977; Witham et al., 2005; Delmelle et al., 2007; Ayris et al., 2013; Schmauss and Keppler, 2014; Maters et al., 2017; Martin, 2018; Urupina et al., 2019).

Line 64: As well as the physical proximity of S-bearing gases/aerosols and ash particles - these species cannot interact if they are spatially separated following an eruption.

Line 65: Also SO₂ gas and ash particles.

Line 79: There seems to be missing a transition here from talking about SO₂ lifetime to talking about aerosols scattering light, for example explaining to readers that these (sulfate/sulfuric acid) aerosols derive from the oxidation of volcanic SO₂.

Lines 90-91: Please specify 'initial' uptake efficiencies here as these can differ greatly from steady state uptake efficiencies.

Lines 92-94: I do not see a purpose or justification for including this statement. The SO₂ gas molecules only interact immediately with the geometric surface area of the solid sample; being unable to access all external and internal (BET) surface area of the sample over the time scale of the initial uptake. Therefore, it is not valid to adjust the initial uptake efficiency values from Maters et al. according to the BET surface area, and expressing the data in this way here risks the values being used erroneously in future studies.

Line 96: Please reiterate the factors being referred to here (e.g., OH abundance, H₂O amount, injection altitude, etc.) since it is not immediately clear.

Lines 96-97: Please reword this line. As written, it suggests that you conduct various spacecraft observations in this study. Also, the simulations are not conducted in Figure 1, but rather the results are presented in Figure 1.

Line 99: Please insert 'satellite' or 'spacecraft' observation for clarity.

Line 101: Please define 'DU'.

Lines 102-103: "We also shows the uncertainties to apply..." - this sentence is unclear and only understandable after reading the explanation in the Figure 1 caption. Please clarify and also specify that it refers to the SO₂onAsh data.

Line 103: Please acknowledge here that a significant amount of the green shaded area (the uncertainty in the SO₂onAsh data) overlaps with the 0.2Tg and Base case, and so it cannot be definitively concluded that the SO₂onAsh scenario alone explains the SO₂ decline observed by satellite.

Line 106: What does 'both' refer to here?

Line 109: Missing the word 'that' between 'report' and 'the SO₂'?

Line 112: Please avoid contractions (replace 'doesn't' with 'does not').

Line 118: I believe this should be '26' instead of '22' days?

Line 121: If 'H₂O injection causes no significant reduction' - then why does the LrgH₂O simulation shorten SO₂ lifetime by 4 days (from 22 to 18; Table S1)?

Line 122: Please insert 'satellite' or 'spacecraft' observations for clarity.

Line 136: I believe the word 'of' between 'because' and 'the particle surface area' should be removed?

Line 140: Note that '2.5 g/cm³' is written here but '2.3 g/cm³' is quoted elsewhere. Is this a typo?

Line 140: 'it appears that the Kelut particles are very similar to pumice' - is there any evidence that this is actually the case, or is it only speculation related to the assumed ash density of 0.5

g/cm³? Please provide a reference for pumice density. Could other features such as ash aggregates account for a low density? Could other properties such as non-spherical ash shapes account for persistence in the stratosphere?

Line 142: I believe this should be 'underestimates' not 'overestimates'?

Figure 2: Why is the SO₂onAsh case not shown in the top panel, even if it is almost identical with the Base case? Also, the legend labels can be improved in terms of clarity and consistency. For example, the 'bck alt' in the top panel legend should specify that this corresponds to the Base case. The mixed use of 'bck' and 'alt' and uppercase and lowercase across both panel legends adds to the confusion.

Line 238: There does not appear to be any 'grey' in Figure 3 - should this be 'green' instead?

Line 256: There does not appear to be any 'grey' in Figure 3 - should this be 'green' instead?

Lines 281-291: The results in Figure 5b are not mentioned at all in this section yet they show some interesting trends. For example, it appears that SO₂ reaction on ash produces sulfate faster than SO₂ oxidation to sulfate in the gas phase. Please include some discussion of Figure 5b findings in the text.

Lines 304-305: 'The majority of sulfate is in the form of pure sulfate rather than the mixed sulfate after 20 days of the eruption (6f)' - does this mean that the mixed sulfate formed/taken up on ash falls out before 20 days, and then the pure sulfate originating from gas-phase oxidation of SO₂ dominates after 20 days? Similar to the comment above, this is an interesting result worth remarking on.

Lines 328-329: This sentence is confusing as written. It is not the 'volcanic eruptions' that 'persisted more than 5 months' but rather the ash particles from these eruptions in the stratosphere.

Lines 336-337: Please provide references here to clarify that it has not been 'shown by laboratory experiments' in the present study.

Lines 339-340: This sentence is unclear, by 'heterogeneous reactions' do you mean 'SO₂/ash' reactions? Also, 'favors' should be 'favor'.

Line 342-343: Perhaps it is implied but it would be helpful to state explicitly that this SO₂ conversion could potentially reflect a contribution from SO₂ reaction on ash.

Line 409: Please list the gases in (typical) order of abundance: H₂O, CO₂, SO₂, HCl, H₂S

Table S1: I think the *** in the LrgOH row would better be placed after 'LrgOH' since it does not refer to the H₂O injection amount (0.26 Tg) but rather relates to the OH photolysis rate.

Line 464: Please avoid contractions (replace 'didn't' with 'did not').

References

Ayris, P. M., Lee, A. F., Wilson, K., Kueppers, U., Dingwell, D. B. and Delmelle, P. (2013) SO₂ sequestration in large volcanic eruptions: high-temperature scavenging by tephra. *Geochimica et Cosmochimica Acta*, 110, 58-69.

Delmelle, P., Lambert, M., Dufrêne, Y., Gerin, P. and Óskarsson, N. (2007) Gas/aerosol-ash interaction in volcanic plumes: New insights from surface analysis of fine ash particles. *Earth and Planetary Science Letters*, 259, 159-170.

Galeazzo, T., Bekki, S., Martin, E., Savarino, J. and Arnold, S. R. (2018) Photochemical box modelling of volcanic SO₂ oxidation: isotopic constraints. *Atmospheric Chemistry and Physics*, 18, 17909-17931.

Martin, E. (2018) Volcanic Plume Impact on the Atmosphere and Climate: O- and S-Isotope Insight into Sulfate Aerosol Formation. *Geosciences*, 8, 198.

Maters, E. C., Delmelle, P., Rossi, M. J. and Ayris, P. M. (2017) Reactive uptake of sulfur dioxide and ozone on volcanic glass and ash at ambient temperature, *Journal of Geophysical Research*, 122, 10077-10088.

Rose, W. I. (1977) Scavenging of volcanic aerosol by ash: atmospheric and volcanological implications. *Geology*, 5(10), 621-624.

Schmauss, D. and Keppler, H. (2014) Adsorption of sulfur dioxide on volcanic ashes. *American Mineralogist*, 99, 1085-1094.

Urupina, D., Lasne, J., Romanias, M. N., Thiery, V., Dagsson-Waldhauserova, P. and Thevent, F. (2019) Uptake and surface chemistry of SO₂ on natural volcanic dusts. *Atmospheric Environment*, 217, 116942.

Witham, C. S., Oppenheimer, C. and Horwell, C. J. (2005) Volcanic ash-leachates: a review and recommendations for sampling methods. *Journal of Volcanology and Geothermal Research*, 141, 299-326.

Reviewer #2:

Remarks to the Author:

Persisting Volcanic Ash Particles Impact Stratospheric Extinction and SO₂ Chemistry" by Zhu et al

Zhu et al determine the impact of ultra-fine volcanic ash on the lifetime of SO₂ and sulfate after the eruption of Mt Kelut in 2014. The results are new and of strong interest for scientist working on stratospheric sulfate and the impact of volcanic eruptions on climate. Currently models show very different results when simulating the evolution of a volcanic cloud. This paper highlights the importance of the early face of the particle evolution after the eruption. They describe a process, heterogeneous reactions on ash, that has not been taken into account in previous studies.

The paper describes an important new process. But it includes also many aspects to get this results and takes the reader into the jungle of many simulations along the path to get this result. Partly the text reads more like a model development paper.

Finding the red line, especially in some paragraphs is not always easy, also, because the reader has to jump from one figure to the supplements, and back to the initial text which is not helpful for the flow.

My major concern is related to the injection area. The authors describe different experiments and steps to lower the SO₂ lifetime, before they describe the results with heterogeneous reactions on ash. Injecting far too high in the stratosphere, they do not simulate the observed transport pattern. To overcome this problem, not only the injection altitude is reduced, also the injection area of the volcanic emissions is strongly enlarged.

For my point of view this is a kind of tuning which is difficult and might not be necessary. The results of the Base simulation do not fit better to the OMI data than the 0.2 Tg simulation (Fig S2). Even the the altitude of the injection is decreased, the

authors miss the combination of low injection height (17 - 19 km) and the injection at the location of the volcano into a single grid box, which would reflect reality much better. This should be done and be commented carefully in case of less good results.

Does your result depend stronger on transport and dispersion or on reaction of SO₂ with ash? In case you can not get a comparable lifetime with heterogeneous reactions on ash and one grid box injection, your hypothesis might be wrong.

The discussion chapter reads like it belongs to another article. To my feeling, this is caused by the fact that you do not say clearly what you are talking about. The discussion also misses a critical discussion of possible limitations of the study and the consequences for climate impact. I expected also a discussion of the importance of your finding for smaller and larger eruptions.

L 45: Reference?

L34 and 49: Define volcanic aerosol, either ash or sulfate or both together

L 60: Overlooked sounds strange

L79 Aerosols scatter.....: How is this related to different SO₂ lifetime in small and large eruptions?

Line 97-99: dashed lines and solid lines in the same color refer to the same simulations?

L103 I don't understand the meaning of the green shaded area as it is described here. It becomes clear later in the text or by reading the caption. Sort this better.

L 113pp Base case: Why do you change 3 things at once?

L118: It seems always wrong to me when one has to increase the injection area. In this case the model does not simulate the transport well. It would be at least necessary to give a physical reason. OMI results show a smaller coverage of the volcanic cloud 14 hours after the eruption than your assumption. Fig S2 shows too low values in Base. 0.2Tg looks better. (See above)

L153: You may discuss the green shaded area here.

L180: Reference still Vernier?

In case volcanic aerosol and ash separate as you describe, how can they react in the way explained in Sect 1? Again, the large injection area is difficult. You need to discuss the role of the injection height of ash and SO₂, e.g. in the discussions.

L191 - 193: a coarser grid results in more vertical diffusion when particles sediment. This would cause a shorter lifetime than in a fine vertical grid. Do you have stronger gradients and less mixing in mind with this comment? If yes, say so.

L200: Where in supplement? Give numbers to the sections. It is very disturbing to switch continuously between the text and the supplement and additionally, to search for the right page in the supplement.

L205 40 days is long for ash. The measurements increase with time. This is different to your results. Please comment.

Fig 2: I recommend to add an error range for the dashed line. Vertical lines not explained in the caption.

L 244: It would help the reader if you name the variable and do not only give the value.

Fig 3: ..black and green....

l329: .. persisted more...: The eruption or the volcanic cloud? Both could theoretically be possible in this sentence.

L330 to 332: this sentence is very general and comes a bit out of the blue. Do you want to say that ash should not be neglected?

L334 - 336: This statement is puzzling. May be: when the heterogeneous reactions on ash are not taken into account?

L 341: Do you talk about SO₂ plus ash or SO₂ oxidation? A lot of guessing is necessary to understand this discussion.

L451: Why? This conclusion is not clear to me. The volcanic cloud is dense in the beginning and

spreads over time. This increases uncertainty as more areas are effected by detection limit.

L478: Do you nudge meteorological variables?

Fig S1: Impossible to read numbers in Zhu et al determine the impact of ultra-fine volcanic ash on the lifetime of SO₂

and sulfate after the eruption of Mt Kelut in 2014. The results are new and of strong interest for scientist working on stratospheric sulfate and the impact of volcanic eruptions on climate. Currently models show very different results when simulating the evolution of a volcanic cloud. This paper highlights the importance of the early face of the particle evolution after the eruption. They describe a process, heterogeneous reactions on ash, that has not been taken into account in previous studies.

The paper describes an important new process. But it includes also many aspects to get this results and takes the reader into the jungle of many simulations along the path to get this result. Partly the text reads more like a model development paper.

Finding the red line, especially in some paragraphs is not always easy, also, because the reader has to jump from one figure to the supplements, and back to the initial text which is not helpful for the flow.

My major concern is related to the injection area. The authors describe different experiments and steps to lower the SO₂ lifetime, before they describe the results with heterogeneous reactions on ash. Injecting far too high in the stratosphere, they do not simulate the observed transport pattern. To overcome this problem, not only the injection altitude is reduced, also the injection area of the volcanic emissions is strongly enlarged.

For my point of view this is a kind of tuning which is difficult and might not be necessary. The results of the Base simulation do not fit better to the OMI data than the 0.2 Tg simulation (Fig S2). Even the the altitude of the injection is decreased, the authors miss the combination of low injection height (17 - 19 km) and the injection at the location of the volcano into a single grid box, which would reflect reality much better. This should be done an be commented carefully in case of less good results.

Does your result depend stronger on transport and dispersion or on reaction of SO₂ with ash? In case you can not get a comparable lifetime with heterogeneous reactions on ash and one grid box injection, your hypothesis might be wrong.

The discussion chapter reads like it belongs to another article. To my feeling, this is caused by the fact that you do not say clearly what you are talking about. The discussion also misses a critical discussion of possible limitations of the study and the consequences for climate impact. I expected also a discussion of the importance of your finding for smaller and larger eruptions.

L 45: Reference?

L34 and 49: Define volcanic aerosol, either ash or sulfate or both together

L 60: Overlooked sounds strange

L79 Aerosols scatter.....: How is this related to different SO₂ lifetime in small and large eruptions?

Line 97-99: dashed lines and solid lines in the same color refer to the same simulations?

L103 I don't understand the meaning of the green shaded area as it is described here. It becomes clear later in the text or by reading the caption. Sort this better.

L 113pp Base case: Why do you change 3 things at once?

L118: It seems always wrong to me when one has to increase the injection area. In this case the model does not simulate the transport well. It would be at least necessary to give a physical

reason. OMI results show a smaller coverage of the volcanic cloud 14 hours after the eruption than your assumption. Fig S2 shows too low values in Base. 0.2Tg looks better. (See above)

L153: You may discuss the green shaded area here.

L180: Reference still Vernier?

In case volcanic aerosol and ash separate as you describe, how can they react in the way explained in Sect 1? Again, the large injection area is difficult. You need to discuss the role of the injection height of ash and SO₂, e.g. in the discussions.

L191 - 193: a coarser grid results in more vertical diffusion when particles sediment. This would cause a shorter lifetime than in a fine vertical grid. Do you have stronger gradients and less mixing in mind with this comment? If yes, say so.

L200: Where in supplement? Give numbers to the sections. It is very disturbing to switch continuously between the text and the supplement and additionally, to search for the right page in the supplement.

L205 40 days is long for ash. The measurements increase with time. This is different to your results. Please comment.

Fig 2: I recommend to add an error range for the dashed line. Vertical lines not explained in the caption.

L 244: It would help the reader if you name the variable and do not only give the value.

Fig 3: ..black and green....

l329: .. persisted more...: The eruption or the volcanic cloud? Both could theoretically be possible in this sentence.

L330 to 332: this sentence is very general and comes a bit out of the blue. Do you want to say that ash should not be neglected?

L334 - 336: This statement is puzzling. May be: when the heterogeneous reactions on ash are not taken into account?

L 341: Do you talk about SO₂ plus ash or SO₂ oxidation? A lot of guessing is necessary to understand this discussion.

L451: Why? This conclusion is not clear to me. The volcanic cloud is dense in the beginning and spreads over time. This increases uncertainty as more areas are effected by detection limit.

L478: Do you nudge meteorological variables?

Fig S1:Impossible to read numbers in bottom figure!

Fig S4:Impossible to read numbers!

We appreciate your time and effort to review the manuscript. Your thoughtful comments helped us to improve the quality of the manuscript. Please view our detailed response below for each question. The question is noted in black, and our response in blue.

Reviewer #1 (Remarks to the Author):

SUMMARY

This modelling study addresses a very interesting question regarding the impact of volcanic ash on the stratospheric S burden, using the 2014 Mt. Kelut eruption as a case study. Comparing the output of various model simulations with satellite observations following this eruption, the authors conclude that including SO₂ reaction on ash particles produces the best match to real observations of stratospheric SO₂ lifetime. This is in addition to the interaction of ash particles with SO₄²⁻ and H₂SO₄ before fallout. Further, they suggest that a very low ash density of 0.5 g/cm³ can explain the ash persistence in the stratosphere, based on comparison to aircraft particle size distribution observations. The study represents a novel assessment of the importance of volcanic S/ash interactions in the stratosphere, and could ultimately represent a high impact publication contributing to new knowledge in this field.

However, as it currently stands, it seems that incorrect values of SO₂ uptake on ash are used in the model simulations, several key SO₂ oxidation pathways in the atmosphere are not included in the model or even acknowledged, the various S species and their interaction with ash are not always expressed in a clear or consistent manner, and a difference in ash shape (from spherical) is not considered as a factor potentially contributing to ash persistence in the stratosphere.

Overall these issues, in addition to numerous minor points, lead me to recommend rejection of this manuscript. However, if the authors are able to confirm their findings using the correct SO₂ uptake values, I would encourage them to consider resubmission.

MAJOR COMMENTS

- The uptake efficiency and saturation coverage values used in the model simulations (lines 144-147), based on the laboratory data of Maters et al. (2017), are off by 2 orders of magnitude in opposing directions. The authors correctly assume that only 'the geometric surface is available for initial uptake' (line 146). However, they then use an initial uptake efficiency value on the order of 10⁻⁴ normalized to the BET surface area, when in fact they should use a value on the order of 10⁻² normalized to the geometric surface area (as reported in lines 90-92). The authors then incorrectly 'assume the geometric surface area' in considering the saturation coverage (line 147), when in fact SO₂ should be able to access both external and internal (BET) surfaces over the saturation timescale. Hence, they use a saturation coverage value on the order of 10¹⁵

cm² normalized to the geometric surface area, when in fact they should use a value on the order of 10¹³ cm² normalized to the BET surface area (as reported in lines 94-95 and 148-150). Thus, the claim (line 151) that the 'assumptions are similar to those used in Maters et al. for volcanic ash' is incorrect, and unfortunately an initial uptake efficiency 2 orders of magnitude too low and a saturation coverage 2 orders of magnitude too high are used in the model simulations. The correct values will need to be adopted to check if the revised model simulation results allow the same conclusions to be drawn regarding the importance of SO₂ reaction on ash in this study.

Our study is an independent in situ determination of the reaction efficiency, rather than attempting to use values from laboratory study. Our measurement is somewhat ambiguous because the area of an individual volcanic particle is not well known. We find, based on the amount of SO₂ in the gas phase over time, and assuming the spherical surface area of the particles is the true collision area, that a reaction efficiency of about 9x10⁻⁴ produces the best agreement with the observations. If in fact, the area is larger than that of a sphere the reaction efficiency would have to be reduced. Another issue is determining when the particle surface coverage becomes saturated, so that uptake of SO₂ stops. We performed several simulations to determine the saturation coverage and found assuming 13% of SO₂ by mass as a saturation coverage on particles with a composition of ash mixed with sulfate provides good agreement with the observed SO₂ lifetime. Again, one cannot directly compare our surface coverage with lab data since we use the surface area of a sphere as the true area. In order to better explain these ideas, we altered the text as described below. We also changed all relevant figures in the paper to update the SO₂onAsh case. We found these changes do not affect any of our conclusions.

Here are the content changes:

L236-274: "In addition to the SO₂ gas-phase reaction with OH, laboratory experiments report SO₂ reacting on volcanic ash and mineral dust¹⁵⁻¹⁸ (Supplement 2). In general, the laboratory data show a rapid initial SO₂ uptake, that saturates over time. We are able to measure the uptake efficiency using the rate of decline of the SO₂ concentration during approximately the first day, and the saturation coverage using the time at which the SO₂ loss stops being controlled by the heterogeneous loss, about 1 day (supplement 2). During this time about 20 % of the initial SO₂ has been removed from the gas phase. We find the uptake efficiency, γ , is about 9x10⁻⁴, and the saturation coverage occurs when about 13% of the mass of the particle is SO₂ from the heterogeneous reactions. These numbers will be useful to climate modelers simulating volcanic clouds, but because we assumed the particles are spherical they are not directly comparable to laboratory studies in which the surface area of the particles is known. The surface area of real volcanic ash is expected to be larger than the spherical assumptions, and the true γ value should be reduced by the amount the true surface area exceeds that of an equal mass sphere. We argue below that the Kelut particles are porous, or non-spherical, or both. For our example of an oblate spheroid particle discussed below the surface area is 1.4 times greater, so the γ value would be about 6x10⁻⁴, but the true surface area is likely larger. Usher et al.¹⁸ reported an uptake efficiency in the range 7.0x10⁻⁵ to 5.1x10⁻⁴ for their mineral samples studied in the laboratory assuming the BET surface area of the particles. Maters et al.¹⁵ used a Knudsen cell to measure the initial uptake efficiency of SO₂ on dicite glass to be 1.3x10⁻² at ~ 250 K assuming the

geometric surface area of the sample holder. Our derived values for Kelut ash assuming geometric surface area of the particles falls between these two laboratory studies. Our surface coverage limit is equivalent to 1.6×10^{16} molecules/cm², which should be scaled down by the actual surface area to compare with laboratory data. As discussed in Supplement 2 the adsorption surface areas can be orders of magnitude higher than the spherical surface area. Several papers in the literature report saturation coverage of SO₂ on the mineral dust or ash ranging from 10^{11} to 5×10^{14} molecules/cm² assuming the BET surface area using different laboratory technologies under different exposure time, pressure, RH and SO₂ concentration^{15-18,25,26}.

Figure 1 displays a case named SO₂onAsh with $\gamma = 9 \times 10^{-4}$ and saturation coverage equivalent to 13% of the SO₂ particle by mass. After saturation, we set the SO₂ uptake efficiency to zero. This choice has the smallest root mean square deviation of the model from the data. Alternative values are explored in Supplement 2. Figure 1 shows the modeled actual SO₂ lifetime, in which all SO₂ in the gas phase is counted and the heterogeneous reaction is included is 17 days (blue dash line). We apply the OMI observational uncertainties on the SO₂onAsh case (the green shaded area). The uncertainties increase quickly and overlap the Ref case and Base case after 10 days, which is consistent with the end time of the OMI observations when the satellite is no longer able to observe SO₂.”

- Excluding the heterogeneous reaction of SO₂ on ash, only the gas-phase reaction of SO₂ with OH is acknowledged and modeled here as an SO₂ oxidation pathway in the volcanic plume and/or in the atmosphere (lines 70-73, 82, 122-133, Table S1, Figure 1), whereas the liquid-phase reactions of SO₂ with H₂O₂, O₃, and O₂ (catalysed by transition metal ions) are ignored despite also being key SO₂ oxidation pathways. These latter reactions might be as important as SO₂ reaction with OH following an eruption, especially as the water-rich plume might provide significant amounts of water locally to the stratosphere, and this should be included in the model simulations here as well as acknowledged with reference to highly relevant recent literature on this topic (e.g., Martin, 2018; Galeazzo et al., 2018).

We agree with you that in the lower altitude part of a volcanic cloud that liquid phase reactions could be important. We haven't included these reactions in the chemistry since our study of SO₂ chemistry for the 2014 Mt. Kelut involves SO₂ observed in the stratosphere. Temperatures involved are around 220K, and the humidity is only a few percent. Therefore, liquid water is not present and even ice is rapidly evaporating. We think it is a good idea to include more volcanic SO₂ chemistry discussion in the Methods section, so it is clearer to people why our simulations omit the SO₂ aqueous phase chemistry.

Line 635: add “Figure 7 also shows the main SO₂ chemistry we are considering in this paper: SO₂ uptake on ash and SO₂ reaction with OH. SO₂ aqueous phase chemistry is also important for tropospheric volcanic SO₂ oxidation, but is not included here since liquid water is not expected in the stratosphere^{14,42}, and since we focus on SO₂ and stratospheric volcanic cloud evolution days and months after the eruption. However, including the effects of these tropospheric reactions on the volcanic plume would be valuable to better understand the total sulfate deposition on ash, some of occurs before the ash enters the stratosphere⁹⁻¹³.”

Line 84: "Finally, we show that there is significant removal of sulfur (S) on falling ash. Such removal has been observed in the troposphere near volcanic vents possibly due to adsorption of sulfur gases on the ash⁹⁻¹⁴. However, SO₂ removal via heterogeneous reactions on ash in the stratosphere has been ignored in climate models, even though it is recognized in laboratory studies¹⁵⁻¹⁸."

Line 111: modified to "Generally, in the stratosphere, it has been thought that the SO₂ lifetime is determined by its reaction rate with OH."

- There seems to be confusion in the way that various S-bearing species as well as their interaction with volcanic ash are expressed throughout the manuscript, which unfortunately translates to the reader a lack of clarity of the processes being discussed. The authors could better distinguish the different S chemistry pathways and the role of ash in this context. For example, various terminology is used to describe SO₄²⁻/H₂SO₄ interaction with ash including 'growth on', 'coagulation on', and 'nucleation on' (e.g., lines 303-304, 381-383), which refer to different processes but this is not well explained. Including Figure S9 in the manuscript instead of the supplement could partly help here. Moreover, the uptake of SO₄²⁻/H₂SO₄ on ash surfaces (by whatever process) is distinct from the formation of SO₄²⁻/H₂SO₄ on ash surfaces following the uptake and oxidation of SO₂ gas. Clarifying this distinction is important because it means that, even if volcanic SO₂ gas has not yet been oxidized to SO₄²⁻/H₂SO₄ in the atmosphere (by gas- or liquid-phase reactions), ash is still able to act as a sink for volcanic S by reacting with the SO₂ gas. This comes across in some parts of the text but not others. For instance, why is SO₂ gas and ash reaction not included in Figure S9?

Figure S9 is now Figure 7.

Figure 7 now includes SO₂ gas and its processes to interact with ash and convert to H₂SO₄ (a.k.a. processes 6. And 7.)

We also include a schematic diagram in addition to the model processes in Figure 7.

Additionally, there is inconsistency throughout the text in the use of the chemical compound names versus their chemical formulas. For example, 'sulfur dioxide' is never defined and only 'SO₂' is used, 'sulfuric acid' and 'H₂SO₄' are interchanged throughout, and 'sulfate' is used and never defined as 'SO₄²⁻'.

We defined them at the first place they appear.

Line 28: "the sulfate aerosol (a.k.a. sulfuric acid aerosol or SO₄²⁻)".

Line 31: "sulfuric acid (H₂SO₄)".

Line 32: "sulfur dioxide (SO₂)".

Line 82: "Hydroxide (OH)".

Line 84: "Sulfur (S)".

Relatedly:

Lines 67-69: This sentence is confusing as written. First, 'the longer SO₂ takes to convert to H₂SO₄' is probably more accurate as 'the longer it takes SO₂ to be converted to H₂SO₄'. Second, it is not entirely clear how 'the less H₂SO₄ vapor is available to grow onto ash' and 'the fewer sulfuric acid particles are available to coagulate on ash'

particles' differ, please explain the distinction between these two modes of H₂SO₄ uptake on ash.

Line 94: add "Ash can interact with sulfuric acid gas and aerosol through several well-known microphysical processes (Methods, Figure 7): H₂SO₄ gas heterogeneously nucleates on ash particles and then grows; pure sulfuric acid particles formed by homogeneous nucleation and can then coagulate with ash particles and mixed ash/sulfate particles. The longer it takes SO₂ to be converted to H₂SO₄"

In addition, the entire statement seems to neglect that even if SO₂ gas has not yet been converted to SO₄²⁻/H₂SO₄ (by gas- and liquid-phase oxidation pathways), ash particles could still play a role in capturing volcanic S by reacting directly with SO₂ gas before falling out of the stratosphere.

L109: add "Heterogeneous uptake of SO₂ on ash can remove SO₂ directly without requiring SO₂ conversion to H₂SO₄ vapor (Methods, Figure 7)."

Lines 277-281: What is meant by 'growth of sulfuric acid gas on ash'? Ash can also take up volcanic S by reaction with SO₂ gas but this is not mentioned in the opening line. However, later the 'SO₂onAsh case' is then included in what the authors refer to as 'sulfate/ash interactions' investigated in model simulations, but as written this does not encompass SO₂/ash interaction. The broader 'sulfur/ash interactions' could be more accurate.

L459: Change to "Volcanic ash can accumulate sulfur species by uptaking sulfuric acid gas, sulfuric acid aerosols and SO₂ gas."

L461: change to "sulfur/ash".

Lines 283: 'sulfate cannot be effectively removed by ash when the SO₂ lifetime is long' - to avoid any misunderstanding please state explicitly that this is when reaction between SO₂ and ash is neglected. Also, is it just 'sulfate' or also 'sulfuric acid'?

L464: Change to "First, neither the Noashemission case nor the Base case, with a sulfur residence time of 22 days, consider SO₂ reactions on ash. The difference of S burden (Figure 5a) between them is minor indicating the sulfate and sulfuric acid gas is not effectively removed by ash since the SO₂ lifetime is long compared with the super-micron ash removal time. As we include the SO₂ reacting on ash surfaces, the sulfate produced by reactions on the ash stays with the ash particle and is removed effectively."

- It appears that only variation in ash density is considered to explain the Mt. Kelut ash persistence in the stratosphere (lines 50-53, 138-140, 325-327, Table S2), whereas ash shape could also be a contributing factor and variation in this physical property should similarly be investigated in the model simulations and compared with the ATTREX observations. In other words, alongside of challenging the assumption that ash has a density of ~2.3 g/cm³ (as done in this study), it would be as valid to challenge the assumption that ash is a spherical particle (line 239, Table S2). Without doing this, it seems that various possibilities to explain the airborne ash lifetime have not been fully explored here, and the conclusion that a low ash density must be responsible is seriously weakened.

We agree that the falling velocity of ash can be reduced due to non-spherical shape of ash particles.

We ran a model test to see how oblate shapes affect the fall velocity. The figure below shows that assuming an oblate particle with a/b ratio of 0.3 doesn't change the fall velocity very much (the green line almost overlaps the yellow line). Only very highly oblate particles (a/b ratio = 0.1) slightly slow down the fall velocity. For a 1 μm particle, a spherical particle falls down 0.02 km per day while the oblate particle falls down 0.014 km per day. In contrast, changing the density from 2.3 g/cm³ to 0.6 g/cm³ result in a change of fall velocity from 0.07 to 0.02 km per day.

Also, pumice particles usually have very non-spherical shapes with holes in them.

Line 72: add "Non-spherical shapes can also reduce the fall velocity."

Line 394: "Our model tests (not shown) indicate particles with length/diameter ratios of 0.3 only reduce the fall velocity by 8% compared with the equivalent spherical particle, while highly oblate particles with length/diameter ratio of 0.1 can reduce the fall velocity by 30%."

Line 421: "Note that, in reality, ash particles are usually non-spherical with porous structure. Our conclusion does not mean that the ash density is exactly 0.5 g/cm³. The particles are non-spherical and porous, or could be aggregates, with fall velocities equivalent to spherical particles with densities of 0.5 g/cm³."

Line 532: add "However, the particle shape may also impact the fall velocity. We conclude that density and shape combine to produce a fall velocity equivalent to spherical particles with a density of 0.5 g/cm³, but we cannot determine the relative importance of shape and density."

Relatedly:

Lines 55-56: Model simulations are performed which show that an ash density of 0.5 g/cm³ delivers the best match to particle size distributions observed (ATTREX flight measurements) after the Mt. Kelut eruption. It is not entirely clear if variation in only this one physical property (ash density) is tested to explain the persisting ash in the stratosphere. Did the authors consider other physical properties such as ash shape

(despite lines 232-233 mentioning that a shape change from spherical to oblate could reduce ash fall speed)? Lines 429-431 state “we vary the total mass of ash, the percentage of the particle mass in each of the three size modes, the ash particle density, the latitude range of the injection and the geographic location of the injection to best agree with CALIPSO satellite and ATTREX aircraft data.” Maybe this is what the authors refer to when claiming to investigate different physical processes?

Line 79: change to “We analyze the ash physical properties (particle density, particle size distribution, injection location, etc.) and determine the temporal contribution of volcanic ash to the stratospheric volcanic aerosol layer.”

Line 394: “Our model tests (not shown) indicate particles with length/diameter ratios of 0.3 only reduce the fall velocity by 8% compared with the equivalent spherical particle, while highly oblate particles with length/diameter ratio of 0.1 can reduce the fall velocity by 30%.”

- Lines 19-20: Is the claim that ash removes 50% more S correct? Lines 286-288 state that ‘In two months, the stratospheric S burden is reduced to 51% of its original value in the case with the heterogeneous reaction [on ash] compared to 68% with the Noashemission case and the Base case.’ To me this implies that the addition of S chemistry on ash only removes 17% (not 50%) more volcanic S from the stratosphere compared to if these heterogeneous reactions are ignored.

This phrase might be ambiguous. We referred to the amount removed being 50% larger with heterogeneous chemistry than without. We did not mean the total Sulfur was reduced by 50%.

Line 471: Add “The amount of S removed is 57% larger with heterogeneous chemistry than without.”

Line 21: “About 57% more volcanic sulfur is removed from the stratosphere in 2 months with the SO₂ heterogeneous chemistry on ash particles than without.”

MINOR COMMENTS

Line 1: The title could be improved. First, the term ‘stratospheric extinction’ does not appear anywhere in the manuscript, and its meaning here is unclear. Second, this study does not actually provide new knowledge on the role of ash in ‘SO₂ chemistry’, but incorporates existing data on ash-SO₂ reaction in model simulations to reveal an impact of ash on ‘SO₂ lifetime’ in the stratosphere.

Change the title: “**Persisting Volcanic Ash Particles impact stratospheric SO₂ lifetime and aerosol optical properties**”.

Line 25-27: It seems misleading to include ‘volcanic minerals, referred to as ash’ here in describing the volcanic aerosols that ‘have altered climate throughout Earth’s history’. Traditionally, the stratospheric volcanic aerosols considered to affect climate (by processes mentioned in lines 27-29) refer to liquid sulfuric acid/sulfate aerosols, not to ash particles. Further, the term ‘volcanic mineral’ is inaccurate since volcanic glass (which is non-crystalline) is typically a major component of ash.

Move the second sentence "The major constituents of volcanic aerosols are sulfuric acid (H₂SO₄), originating from sulfur dioxide (SO₂) injections, and volcanic rocks, referred to as ash." to L31.

Line 32: change "volcanic minerals" to "volcanic rocks".

Line 31: 'large, high-density' is vague - please clarify/define.

L34: add "(over 2 g/cm³)".

Line 34: 'relatively small' is vague - please clarify/define (e.g., using the Volcanic Explosivity Index).

L54: add "with Volcanic Explosivity Index of 4".

Line 43: 'volcanic ash was observed' - please insert 'in the stratosphere' to clarify the context.

L62: fixed.

Lines 45-46: Please provide a reference related to the 'Mt. St. Helens in 1980' case. Also, please clarify what specifically is meant by 'available ash observations' (e.g., 'by satellite'), since ash is often 'observed' in general during or after an explosive eruption.

L64: reference added.

L65: modify to "ash size distributions are observed soon after the eruptions".

Line 50: Please provide a reference for the value of 2.3 g/cm³. Typically a value closer to 2.5 or 2.6 g/cm³ is reported, and 2.5 g/cm³ is also quoted in this study on line 140.

L69: add citation [Durant et al., 2009]. This paper categorizes the volcanic particles into 3 types: pumice of 0.6 g/cm³, ice of 1 g/cm³, and glass of 2.3 g/cm³, which is not from observations, but gives an approximate range of density for each particle type.

Other related models use densities of 2.4 g/cm³ [Niemeier et al., 2009], 2.5 g/cm³ [Vernier et al., 2016] and 2.5 g/cm³ [Kristiansen et al., 2015]. So we use "~" here because of this spread.

Line 59: Please insert 'that' between 'show' and 'there is'.

L84: fixed.

Line 60: 'a process previously overlooked' - please clarify the context you refer to (e.g., 'in models'). The notion that airborne volcanic ash/glass can capture SO₂/SO₄²⁻/H₂SO₄ is well recognized in the literature (e.g., Rose, 1977; Witham et al., 2005; Delmelle et al., 2007; Ayris et al., 2013; Schmauss and Keppler, 2014; Maters et al., 2017; Martin, 2018; Urupina et al., 2019).

L84: Add "Such removal has been observed in the troposphere near volcanic vents possibly due to adsorption of sulfur gases on the ash⁹⁻¹⁴. However, SO₂ removal via heterogeneous reactions on ash in the stratosphere has been ignored in climate models, even though it is recognized in laboratory studies¹⁵⁻¹⁸."

Line 64: As well as the physical proximity of S-bearing gases/aerosols and ash particles - these species cannot interact if they are spatially separated following an eruption.

L91: add "as well as the physical proximity of sulfur species and ash particles."

Line 65: Also SO₂ gas and ash particles.

L92: Change "H₂SO₄ vapor" to "sulfur species".

Line 79: There seems to be missing a transition here from talking about SO₂ lifetime to talking about aerosols scattering light, for example explaining to readers that these (sulfate/sulfuric acid) aerosols derive from the oxidation of volcanic SO₂.

The aerosols here are not only volcanic sulfate but also include background aerosol, as well as ash or ice, any particles that can scatter the light.

The logic of this paragraph is to list all the factors we can think of that influence the SO₂+OH reaction rate.

L114: "The factors that impact the SO₂/OH reaction rate include: the injection of H₂O from the volcanic eruption, which can increase the OH concentration; the spreading of the volcanic clouds, which dilutes the SO₂ concentration so OH is less impacted by the reaction with SO₂; as well as light scattering by ash, sulfate, and possibly ice, which increases the path length of light through the atmosphere thereby changing photorates that control the OH production rate."

Lines 90-91: Please specify 'initial' uptake efficiencies here as these can differ greatly from steady state uptake efficiencies.

Line 254: The whole paragraph is rewritten.

Lines 92-94: I do not see a purpose or justification for including this statement. The SO₂ gas molecules only interact immediately with the geometric surface area of the solid sample; being unable to access all external and internal (BET) surface area of the sample over the time scale of the initial uptake. Therefore, it is not valid to adjust the initial uptake efficiency values from Maters et al. according to the BET surface area, and expressing the data in this way here risks the values being used erroneously in future studies.

This part is deleted. Refer to the answer to major comment question 1.

Line 96: Please reiterate the factors being referred to here (e.g., OH abundance, H₂O amount, injection altitude, etc.) since it is not immediately clear.

L125: modified to "To explore the influence of various factors that might alter the SO₂ oxidation rate in our model, we show in Figure 1 several simulations (lines) , as well as various spacecraft observations (symbols)."

Lines 96-97: Please reword this line. As written, it suggests that you conduct various spacecraft observations in this study. Also, the simulations are not conducted in Figure 1, but rather the results are presented in Figure 1.

L125: modified to "To explore the influence of various factors that might alter the SO₂ oxidation rate in our model, we show in Figure 1 several simulations (lines) , as well as various

spacecraft observations (symbols)."

Line 99: Please insert 'satellite' or 'spacecraft' observation for clarity.
Reconstruct the paragraph and this sentence is deleted.

Line 101: Please define 'DU'.

Line 284: "Dobson Unit (DU)"

Lines 102-103: "We also shows the uncertainties to apply..." - this sentence is unclear and only understandable after reading the explanation in the Figure 1 caption. Please clarify and also specify that it refers to the SO₂onAsh data.

Line 282: now reads "The dashed lines and solid lines in the same color refer to the same simulations, but the solid lines show the simulated SO₂ burdens counting only the grid cells where SO₂ is above OMI noise level (0.2 Dobson Unit (DU), while the dashed lines show the simulated total SO₂ burden without accounting for observational bias due to signal to noise ratio."

Line 103: Please acknowledge here that a significant amount of the green shaded area (the uncertainty in the SO₂onAsh data) overlaps with the 0.2Tg and Base case, and so it cannot be definitively concluded that the SO₂onAsh scenario alone explains the SO₂ decline observed by satellite.

Line 271: "We apply the OMI observational uncertainties on the SO₂onAsh case (the green shaded area). The uncertainties increase quickly and overlap the Ref case and Base case after 10 days, which is consistent with the end time of the OMI observations when the satellite is no longer able to observe SO₂."

Line 106: What does 'both' refer to here?

The sentence is deleted.

Line 109: Missing the word 'that' between 'report' and 'the SO₂'?

L164: fixed.

Line 112: Please avoid contractions (replace 'doesn't' with 'does not').

L168: fixed.

Line 118: I believe this should be '26' instead of '22' days?

L176: fixed.

Line 121: If 'H₂O injection causes no significant reduction' - then why does the LrgH₂O simulation shorten SO₂ lifetime by 4 days (from 22 to 18; Table S1)?

LrgH₂O injected an unrealistic amount of water in order to decrease the SO₂ lifetime. This case explains that H₂O is not the solution for shortening SO₂ lifetime. While 'H₂O injection causes no significant reduction' refers to an injection of H₂O similar to the MLS observation, which doesn't cause significant reduction of the SO₂ lifetime.

Line 179: added "the observed H₂O injection".

Line 122: Please insert 'satellite' or 'spacecraft' observations for clarity.

L180: add "satellite".

Line 136: I believe the word 'of' between 'because' and 'the particle surface area' should be removed?

The part is deleted.

Line 140: Note that '2.5 g/cm³' is written here but '2.3 g/cm³' is quoted elsewhere. Is this a typo?

The part is deleted.

Line 140: 'it appears that the Kelut particles are very similar to pumice' - is there any evidence that this is actually the case, or is it only speculation related to the assumed ash density of 0.5 g/cm³? Please provide a reference for pumice density. Could other features such as ash aggregates account for a low density? Could other properties such as non-spherical ash shapes account for persistence in the stratosphere?

The part is deleted.

Line 423: add "The particles are non-spherical and porous, or could be aggregates, with fall velocities equivalent to spherical particles with densities of 0.5 g/cm³."

Line 142: I believe this should be 'underestimates' not 'overestimates'?

This part is deleted. Refer to answers to major comment question 1.

Figure 2: Why is the SO₂onAsh case not shown in the top panel, even if it is almost identical with the Base case? Also, the legend labels can be improved in terms of clarity and consistency. For example, the 'bck alt' in the top panel legend should specify that this corresponds to the Base case. The mixed use of 'bck' and 'alt' and uppercase and lowercase across both panel legends adds to the confusion.

Fixed in Figure 2.

Line 238: There does not appear to be any 'grey' in Figure 3 - should this be 'green' instead?

L401: fixed.

Line 256: There does not appear to be any 'grey' in Figure 3 - should this be 'green' instead?

L406: fixed.

Lines 281-291: The results in Figure 5b are not mentioned at all in this section yet they show some interesting trends. For example, it appears that SO₂ reaction on ash produces sulfate faster than SO₂ oxidation to sulfate in the gas phase. Please include some discussion of Figure 5b findings in the text.

Line 471: add "Figure 5b shows that SO₂ reaction on ash (red) produces sulfate faster than SO₂ oxidation to sulfate in the gas phase (black)."

Lines 304-305: 'The majority of sulfate is in the form of pure sulfate rather than the mixed sulfate after 20 days of the eruption (6f)' - does this mean that the mixed sulfate formed/taken up on ash falls out before 20 days, and then the pure sulfate originating from gas-phase oxidation of SO₂ dominates after 20 days? Similar to the comment above, this is an interesting result worth remarking on.

Line 503: "This dominance of pure sulfate indicates SO₂ oxidation to sulfuric acid in the gas phase followed by nucleation into pure sulfate is the dominant sulfate production processes at this time since the heterogeneous reaction of SO₂ on ash has shut down."

Lines 328-329: This sentence is confusing as written. It is not the 'volcanic eruptions' that 'persisted more than 5 months' but rather the ash particles from these eruptions in the stratosphere.

This sentence is deleted.

Lines 336-337: Please provide references here to clarify that it has not been 'shown by laboratory experiments' in the present study.

L546: Reference added.

Lines 339-340: This sentence is unclear, by 'heterogeneous reactions' do you mean 'SO₂/ash' reactions? Also, 'favors' should be 'favor'.

L568: fixed.

Line 342-343: Perhaps it is implied but it would be helpful to state explicitly that this SO₂ conversion could potentially reflect a contribution from SO₂ reaction on ash.

L568: change the sentence to "The SO₂/ash reactions enhance the sulfur/ash interactions and favor more sulfate removal by ash. While our simulations are focused on a relatively small eruption, Guo et al^{28,29} reported that half of the SO₂ released to the stratosphere in the Pinatubo eruption was converted to sulfate within 3 days of the eruption."

Line 409: Please list the gases in (typical) order of abundance: H₂O, CO₂, SO₂, HCl, H₂S

L670: fixed.

Table S1: I think the *** in the LrgOH row would better be placed after 'LrgOH' since it does not refer to the H₂O injection amount (0.26 Tg) but rather relates to the OH photolysis rate.

Table 1: fixed.

Line 464: Please avoid contractions (replace 'didn't' with 'did not').

L307: fixed.

References

Ayris, P. M., Lee, A. F., Wilson, K., Kueppers, U., Dingwell, D. B. and Delmelle, P. (2013) SO₂ sequestration in large volcanic eruptions: high-temperature scavenging by tephra. *Geochimica et Cosmochimica Acta*, 110, 58-69.

Delmelle, P., Lambert, M., Dufrêne, Y., Gerin, P. and Óskarsson, N. (2007) Gas/aerosol-ash interaction in volcanic plumes: New insights from surface analysis of fine ash particles. *Earth and Planetary Science Letters*, 259, 159-170.

Galeazzo, T., Bekki, S., Martin, E., Savarino, J. and Arnold, S. R. (2018) Photochemical box modelling of volcanic SO₂ oxidation: isotopic constraints. *Atmospheric Chemistry and Physics*, 18, 17909-17931.

Martin, E. (2018) Volcanic Plume Impact on the Atmosphere and Climate: O- and S-Isotope Insight into Sulfate Aerosol Formation. *Geosciences*, 8, 198.

Maters, E. C., Delmelle, P., Rossi, M. J. and Ayris, P. M. (2017) Reactive uptake of sulfur dioxide and ozone on volcanic glass and ash at ambient temperature, *Journal of Geophysical Research*, 122, 10077-10088.

Rose, W. I. (1977) Scavenging of volcanic aerosol by ash: atmospheric and volcanological implications. *Geology*, 5(10), 621-624.

Schmauss, D. and Keppler, H. (2014) Adsorption of sulfur dioxide on volcanic ashes. *American Mineralogist*, 99, 1085-1094.

Urupina, D., Lasne, J., Romanias, M. N., Thiery, V., Dagsson-Waldhauserova, P. and Thevent, F. (2019) Uptake and surface chemistry of SO₂ on natural volcanic dusts. *Atmospheric Environment*, 217, 116942.

Witham, C. S., Oppenheimer, C. and Horwell, C. J. (2005) Volcanic ash-leachates: a review and recommendations for sampling methods. *Journal of Volcanology and Geothermal Research*, 141, 299-326.

We really appreciate your time and effort to review the manuscript. Your thoughtful comments helped us to improve the quality of the manuscript. Please view our detailed response below for each question. The question is noted in black, and our response in blue.

Reviewer #2 (Remarks to the Author):

Persisting Volcanic Ash Particles Impact Stratospheric Extinction and SO₂ Chemistry" by Zhu et al

Zhu et al determine the impact of ultra-fine volcanic ash on the lifetime of SO₂ and sulfate after the eruption of Mt Kelut in 2014. The results are new and of strong interest for scientist working on stratospheric sulfate and the impact of volcanic eruptions on climate. Currently models show very different results when simulating the evolution of a volcanic cloud. This paper highlights the importance of the early face of the particle evolution after the eruption. They describe a process, heterogeneous reactions on ash, that has not been taken into account in previous studies.

Thank you for the encouraging comments.

The paper describes an important new process. But it includes also many aspects to get this results and takes the reader into the jungle of many simulations along the path to get this result. Partly the text reads more like a model development paper. Finding the red line, especially in some paragraphs is not always easy, also, because the reader has to jump from one figure to the supplements, and back to the initial text which is not helpful for the flow.

We reconstructed the paper quite a bit. We put several figures and tables back in the main text and methods. We moved the content to different places. Hopefully it is clearer this way.

My major concern is related to the injection area. The authors describe different experiments and steps to lower the SO₂ lifetime, before they describe the results with heterogeneous reactions on ash. Injecting far too high in the stratosphere, they do not simulate the observed transport pattern. To overcome this problem, not only the injection altitude is reduced, also the injection area of the volcanic emissions is strongly enlarged.

For my point of view this is a kind of tuning which is difficult and might not be necessary. The results of the Base simulation do not fit better to the OMI data than the 0.2 Tg simulation (Fig S2). Even the altitude of the injection is decreased, the authors miss the combination of low injection height (17 - 19 km) and the injection at the location of the volcano into a single grid box, which would reflect reality much better. This should be done and be commented carefully in case of less good results.

Does your result depend stronger on transport and dispersion or on reaction of SO₂ with ash? In case you cannot get a comparable lifetime with heterogeneous reactions on ash and one grid box injection, your hypothesis might be wrong.

If we inject SO₂ into one box, it will be more concentrated. We ran several test cases to explore these two major concerns from your description: 1. Does it influence the SO₂ lifetime since we dismiss the SO₂ below 0.2 DU; 2. if SO₂ is injected into different grid boxes than ash since they may be vertically separated, does it influence the SO₂ uptake on ash particles and SO₂ lifetime.

According to the test cases, we add explanations in the supplement and discussion:

L825: rewrite the paragraph “We compare the SO₂ distribution from several simulated cases (described in Table 1) and the OMI data (Figure S3). We inject SO₂ into a 10-degree latitude band for the Base case and the SO₂onAsh case. Both cases show similar transport patterns, but the Base case has higher concentration values. On Feb. 20th, OMI shows a peak concentration of 1.5 DU near 50°E, while the Base case value is near 1.2 DU and the SO₂onAsh is near 0.6 DU. The Ref case shows a similar concentration on the first day, but with a narrower longitudinal range and peaking near 100°E on Feb. 20th. Neither of these cases can explain the detailed distribution shown by OMI.

Due to the uncertainty of the SO₂ spreading pattern, we test simulations with and without injecting SO₂ into a 10-degree latitude band to determine if the injection pattern impacts the SO₂ lifetime. We find the SO₂ injection pattern does not affect our conclusion that the uptake of SO₂ on ash is the major control for SO₂ lifetime. We need to tune the saturation coverage or uptake efficiency for each simulation, but the tuning is still within the uncertainties of these parameters. We suggest using a finer resolution model in the first couple of days to help with the chemical component and particle transport patterns in the climate model.”

Discussion L547: “We distribute the initial eruption plume over an extended area in order to capture the observed wind shear and plume transport. However, we find that more confined injections do not impact our conclusion that heterogeneous reactions of SO₂ on ash are important.”

Discussion L575: “Also, climate models with higher resolution can reproduce the spread of volcanic ash and SO₂ better.”

The discussion chapter reads like it belongs to another article. To my feeling, this is caused by the fact that you do not say clearly what you are talking about. The discussion also misses a critical discussion of possible limitations of the study and the consequences for climate impact. I expected also a discussion of the importance of your finding for smaller and larger eruptions.

We rewrote the discussion to make it clearer.

L 45: Reference?

L64: reference Added.

L34 and 49: Define volcanic aerosol, either ash or sulfate or both together

We define the volcanic aerosol in the first paragraph, L31: “The major constituents of volcanic aerosols are sulfuric acid (H₂SO₄), originating from sulfur dioxide (SO₂) injections, and volcanic rocks, referred to as ash.”

L 60: Overlooked sounds strange

L86: rewrite the sentence "However, SO₂ removal via heterogeneous reactions on ash in the stratosphere has been ignored in climate models, even though it is recognized in laboratory studies¹⁵⁻¹⁸."

L79 Aerosols scatter.....: How is this related to different SO₂ lifetime in small and large eruptions?

Line 114: We clarified the role of scattering. "The factors that impact the SO₂/OH reaction rate include: the injection of H₂O from the volcanic eruption, which can increase the OH concentration; the spreading of the volcanic clouds, which dilutes the SO₂ concentration so OH is less impacted by the reaction with SO₂; as well as light scattering by ash, sulfate, and possibly ice, which increases the path length of light through the atmosphere thereby changing photorates that control the OH production rate."

Line 97-99: dashed lines and solid lines in the same color refer to the same simulations? Yes. Line 282: add "The dashed lines and solid lines in the same color refer to the same simulations."

L103 I don't understand the meaning of the green shaded area as it is described here. It becomes clear later in the text or by reading the caption. Sort this better.

We rearrange the paragraph. Move Table S1 to Table 1.

Line 271: Add "We apply the OMI observational uncertainties on the SO₂onAsh case (the green shaded area). The uncertainties increase quickly and overlap the Ref case and Base case after 10 days, which is consistent with the end time of the OMI observations when the satellite is no longer able to observe SO₂."

L113pp Base case: Why do you change 3 things at once?

L175: "We included these three factors together because their effects are each small, in total they shorten the lifetime by 4 days relative to the Mills et al.²⁰ lifetime of 26 days. With three tests tracking the three factors individually, we found the larger geographic injection area for SO₂ contributes 2 days to lifetime reduction, the lower injection height contributes 2 days, and the observed H₂O injection causes no significant reduction."

L118: It seems always wrong to me when one has to increase the injection area. In this case the model does not simulate the transport well. It would be at least necessary to give a physical reason. OMI results show a smaller coverage of the volcanic cloud 14 hours after the eruption than your assumption. Fig S2 shows too low values in Base. 0.2Tg looks better. (See above)

We explained this a little better on line 169: "The Base case explores the effect on the SO₂ lifetime by altering three factors to be closer to the Microwave Limb Sounder (MLS) observed SO₂ and H₂O. First, we used an initial injection into a 10° latitude x 2° longitude band so that we could pick up enough wind shear to reproduce the spreading patterns and concentrations observed by the OMI satellite (Supplement, Figure S1-S3)."

We also rewrite the paragraph in supplement (refer to the answers to the major concern above).

L153: You may discuss the green shaded area here.

We decided to just mention the details of green shaded area in the caption of Figure 1 and Table 1 in supplement to avoid repeating.

L180: Reference still Vernier?

Yes.

In case volcanic aerosol and ash separate as you describe, how can they react in the way explained in Sect 1? Again, the large injection area is difficult. You need to discuss the role of the injection height of ash and SO₂, e.g. in the discussions.

Please refer to the answer to the first question. We add the description of this limitation of the model simulation to the discussion L575: "Also, climate models with higher resolution can reproduce the spread of volcanic ash and SO₂ better."

L191 - 193: a coarser grid results in more vertical diffusion when particles sediment. This would cause a shorter lifetime than in a fine vertical grid. Do you have stronger gradients and less mixing in mind with this comment? If yes, say so.

line 330 to clarify we added "The altitude differences between the simulations and observations in Figure 2 can be caused by the mismatch of the vertical resolution of the model, which is about 1 km in the stratosphere, and the CALIPSO data, which is about 60 meters."

L200: Where in supplement? Give numbers to the sections. It is very disturbing to switch continuously between the text and the supplement and additionally, to search for the right page in the supplement.

L344: Add "Figure S1".

L205 40 days is long for ash. The measurements increase with time. This is different to your results. Please comment.

I think we picked a too narrow latitude range. From Figure S3, you can see the latitudinal width of the layer is totally ~10 degrees and this band floats between ~15S and ~5N. It is hard to have a precise comparison between model and the data right at 8S (with a 2-degree latitudinal range).

Also, by revisiting the data, we found the area from the 0-6S actually has slightly larger backscatter than 8S. Therefore, we plot the 0S - 6S instead.

The new plots (Figure 2) shows the updated zonal average lines for from base, SO₂onAsh and CALIPSO. You can see the CALIPSO zonal average doesn't have an obvious increasing trend anymore (the first point around 22 days is 1.89 and the last point around 62 days is around 1.84), but the trend is very flat (almost no decrease of the backscatter). In contrast, the two simulated backscatter cases decrease showing slowly declining backscatter from 20 to 60 days, but it is still within 20% of the CALIPSO zonal average data (grey shaded area).

Fig 2: I recommend to add an error range for the dashed line. Vertical lines not explained in the caption.

We add the +/-20% shaded area for the CALIPSO zonal average data.

L368: Vertical line explained: "The grey error bars represent the standard deviation of the dotted CALIPSO data. The grey shaded area shows the modeled average values are within 20% of CALIPSO averaged values. The standard deviation of the CALIPSO zonal average values is over a factor of 200%."

L 244: It would help the reader if you name the variable and do not only give the value.
Fig 3: ..black and green....

L405: modified to "Decreasing the particle density from 2.3 g/cm³ (Figure 3c) to 0.5 g/cm³ (Figure 3a) significantly improves the observed (black and green) and simulated (red) particle size distribution agreement around 5 μm".

l329: .. persisted more...: The eruption or the volcanic cloud? Both could theoretically be possible in this sentence.

This sentence is deleted.

L330 to 332: this sentence is very general and comes a bit out of the blue. Do you want to say that ash should not be neglected?

L542: change to "We find that volcanic sulfate can be removed by volcanic ash more rapidly than considered in climate models which have ignored heterogeneous reactions of SO₂ on ash."

L334 - 336: This statement is puzzling. May be: when the heterogeneous reactions on ash are not taken into account?

Good suggestion. L544: Change to "We found that the effect of ash on removing volcanic sulfate in a small volcanic eruption is minor when the uptake of SO₂ on ash is not considered."

L 341: Do you talk about SO₂ plus ash or SO₂ oxidation? A lot of guessing is necessary to understand this discussion.

L568: modify the sentence to "The SO₂/ash reactions enhance the sulfur/ash interactions and favor more sulfate removal by ash. While our simulations are focused on a relatively small eruption, Guo et al^{28,29} reported that half of the SO₂ released to the stratosphere in the Pinatubo eruption was converted to sulfate within 3 days of the eruption."

L451: Why? This conclusion is not clear to me. The volcanic cloud is dense in the beginning and spreads over time. This increases uncertainty as more areas are affected by detection limit.

L925: Modify the sentence to "During the first couple of days, the uncertainty is small (the green shaded area is narrow). Except for the SO₂onash case, all the other model cases cannot reduce the SO₂ to be within the green shaded area, which indicates the need to consider the reaction of SO₂ on ash."

L478: Do you nudge meteorological variables?

L726: add "The model is nudged with the Modern-Era Retrospective Analysis for Research and Applications (MERRA) for temperature, zonal and meridional winds, and surface pressure fields."

Fig S1: Impossible to read numbers in bottom figure!
Figure modified.

Fig S4: Impossible to read numbers!
Figure modified.

Reviewers' Comments:

Reviewer #1:

Remarks to the Author:

see files attached.

My comments are highlighted in yellow in the word document.

Additional comments appear in the pdf file as well.

Reviewer #2:

Remarks to the Author:

Zhu et al determine the impact of ultra-fine volcanic ash on the lifetime of SO₂ and sulfate after the eruption of Mt Kelut in 2014. The results are new and of strong interest for scientist working on stratospheric sulfate and the impact of volcanic eruptions on climate. The paper describes an important new process. The readability of the paper is strongly improved. The text is now quite informative.

My major concern in my first review has been the injection area, raising the question that in case you can not get a comparable lifetime with heterogeneous reactions on ash and one grid box injection, your hypothesis might be wrong. The authors write now that they tested whether their hypothesis holds also with an injection into a grid box instead of a latitude band. Unfortunately, they have not include a figure to prove this into the answer to reviewers. I am willing to trust the authors in this point as my concern is more related to the transport of the model than to the ash processes.

I recommend publication of the paper. The topic is important for better understanding the processes related to the simulation of volcanic eruptions in climate models.

L 244 - 246: I wonder were the problem is when you know the shape pf the particle in laboratory studies and in the model. Please rewrite.

Figures: The font size used in many figures is still quite small.

Figure S1: Use SI units hPa, not mbar.

Thank you for your time and effort to review our paper during this uncertain time. Also thank you for sorting out all the small grammatic errors in the manuscript. We copied the highlighted reviewer's comments below and answered them accordingly. We also followed your comments on the manuscript and fixed all the contents. Please view our detailed response below for each question. The question is noted in black, and our response in blue.

Thanks to the authors for their work to improve the structure and clarity of the text, appearance of the figures, and explanation of the SO₂ uptake coefficient and saturation coverage in the model. Most issues have been well addressed but a few outstanding points are highlighted below and also in the attached manuscript.

- I would expect the true surface area to be higher than the geometric surface area by more than 1.4 times not just because of a shape difference, but also because of other features that contribute to this discrepancy such as surface roughness and/or secondary phases, which can raise specific surface area by an order of magnitude or more (Riley et al., 2003; Delmelle et al., 2005; Paque et al., 2016). Please acknowledge this here.

Line 199: add “because of other features such as surface roughness and/or secondary phases, which can raise specific surface area by one or two orders of magnitude^{18,28–30}”.

- MAJOR: There is evidence that irreversible SO₂ uptake by chemical reaction with ash surfaces occurs up to a monolayer coverage, estimated to be on the order of 5×10^{14} molecules cm⁻², with saturation coverages of no more than ~0.2% SO₂ by mass (Schmauss and Keppler, 2014; Urupina et al., 2019). The very large saturation coverage of 13% SO₂ on ash used in the model here, equivalent to 1×10^{16} molecules cm⁻², would thus reflect dominantly physisorption of SO₂ in multiple layers rather than chemisorption of SO₂ in a monolayer on ash surfaces (Schmauss and Keppler, 2014; Urupina et al., 2019). While it is noted in the text that the saturation SO₂ uptake value can vary by orders of magnitude depending on adopting the geometric surface area (in the model) or the BET surface area (in lab studies), the implications of this discrepancy in terms of what it reflects for the SO₂ uptake mechanism (reversible physisorption versus irreversible chemisorption) is not acknowledged, yet is central to the assumption that ash surfaces are removing SO₂ permanently from the stratosphere. Therefore, to avoid any misunderstanding for readers considering these values for future use, this notion should be highlighted in the text (i.e., laboratory evidence that irreversible SO₂ uptake on ash by chemical reaction saturates at or below a monolayer coverage and not exceeding ~0.2% by mass). It would also be very informative, and is strongly suggested, to run a model simulation using this lower saturation coverage to demonstrate how it impacts the study's conclusion of the importance of heterogeneous reactions for reducing the stratospheric SO₂ lifetime.

To avoid the possible misunderstanding of using mass percentage for the saturation coverage, we recomputed all our simulations using molecules per area [# / cm²] instead of mass weight percentage as the limitation for the saturation coverage.

Line 181: “We find the uptake efficiency, γ , is about 3×10^{-3} , and the saturation coverage is about 3×10^{16} # / cm² assuming a geometric surface area of spherical ash particles. This saturation coverage is equivalent to ~ 13% of SO₂ on mixed particles by mass in our simulation, for the particle size with the largest surface area, ~ 2 μ m.”

Since we assume a geometric surface area for saturation coverage, it needs to be scaled down by 1-2 orders of magnitude to compare with the lab data using the BET surface area. Schmauss and Keppler, (2014) shows evidence to support up to 9×10^{14} #/cm² for saturation coverage after desorption, so we add this citation too. But we don't think physisorption of SO₂ fits our experiment.

Line 206: "Several papers in the literature report saturation coverage of SO₂ on the mineral dust or ash ranging from 10^{11} to 9×10^{14} molecules/cm² assuming the BET surface area using different laboratory technologies under different exposure times, pressures, RH, and SO₂ concentrations^{13,15-18,31,32}. Likewise, if we scale down our saturation coverage limit by 1 or 2 orders of magnitude, the value falls between these laboratory data. Several different combinations of uptake efficiency and saturation coverage can also explain the observed SO₂ burden (Figure S5). Figure S5 indicates that as we increase the uptake efficiency, we can decrease the saturation coverage to fit the observation. Figure S5 suggests a reasonable saturation coverage, assuming spherical particles and the geometric surface area, is in the 10^{16} #/cm² range, and γ is in the 10^{-3} to 10^{-2} range."

- I believe that OH should refer to hydroxyl radical and not to hydroxide (ion) as stated in Line 82. For clarity, please use the correct chemical notation ($\cdot\text{OH}$) when first defined/used.
L80: fixed.
- Since it is now the first figure presented in the manuscript, I believe it should be numbered as Figure 1 not Figure 7.
Figures are renumbered.
- A random mixture of "sulfuric acid" and "H₂SO₄" still appears throughout the manuscript (e.g., Lines 94-99, 110, 459,466, 502, 504, 626 628-630), which reduces readability and may cause confusion. After sulfuric acid is defined as H₂SO₄, only the latter should be used thereafter for consistency and clarity. The same applies to sulfur, which is defined as S, but then continues to appear as sulfur thereafter. Please correct this throughout the text.
Replaced the sulfuric acid gas with H₂SO₄, and replaced sulfur with S throughout the text.
- This is not entirely accurate, fine ash particles are essentially non-porous or have a porosity dominated by macropores (diameters >50 nm; Delmelle et al., 2005). Please revise this statement.
Line 381: modified to "ash particles are usually non-spherical with non-porous structure or with a porosity dominated by macropores (diameters >50 nm)²⁹. Our conclusion is that the non-spherical, porous or aggregate shaped particles have the same fall velocities as spherical particles with densities of 0.5 g/cm³."
- "Injection location" is not an ash physical property. Please revise.
Line 78: delete "Injection location."
- Please specify "volcanic rocks <2 mm in diameter" since this is the size cut-off that defines ash.
Line 30: add "< 2 mm in diameter".

- Is “photorates” a word? Please clarify.
Line 120: change to “photolysis rate” and add citation.
- Okay, but still please specify “initial” uptake coefficients in each case (Lines 873, 875, 881, 887, 888), since it is a key distinction.
Line 201, 780, 784, 790, 792, 798, 805, 806: add “initial”
- So do we actually need SO₂ reaction on ash, or is H₂SO₄ uptake on ash sufficient to explain the sulfate production patterns - I mean given that they are similar in both AshonSO₂ and Base cases (Figures 6 and S9)?
Figures 5 and 6/S9 seem somehow contradictory. The Base case follows the Noashemission case in terms of S burden pattern, yet the Base case follows the AshonSO₂ case in terms of sulfate production patterns - how can this be?
Line 487: Modified to “We also see similar contribution patterns of ash and sulfate to the total backscatter and mixing ratio in the Base case as shown in Figure S10. Note that the sulfate percentage in the mixed particle increases much slower in Figure S10e than in Figure 7e. The slower increase explains why the ash cannot reduce the S burden effectively in the base case.”

- Delmelle, P., Villi ras, F., & Pelletier, M. (2005). Surface area, porosity and water adsorption properties of fine volcanic ash particles. *Bulletin of Volcanology*, 67(2), 160–169.
- Paque, M., Detienne, M., Maters, E., & Delmelle, P. (2016). Smectites and zeolites in ash from the 2010 summit eruption of Eyjafjallaj kull volcano, Iceland. *Bulletin of Volcanology*, 78(9), 61.
- Riley, C. M., Rose, W. I., & Bluth, G. J. S. (2003). Quantitative shape measurements of distal volcanic ash. *Journal of Geophysical Research: Solid Earth*, 108(B10).
<https://doi.org/10.1029/2001JB000818>
- Schmauss, D., & Keppler, H. (2014). Adsorption of sulfur dioxide on volcanic ashes. *American Mineralogist*, 99(5–6), 1085–1094.
- Urupina, D., Lasne, J., Romanias, M., Thiery, V., Dagsson-Waldhauserova, P., & Thevenet, F. (2019). Uptake and surface chemistry of SO₂ on natural volcanic dusts. *Atmospheric Environment*, 217, 116942.

Thank you for your time and effort to review our paper during this uncertain time and thank you for your recognition of our work. Please view our detailed response below for each question. The question is noted in black, and our response in blue.

Reviewer #2 (Remarks to the Author):

Zhu et al determine the impact of ultra-fine volcanic ash on the lifetime of SO₂ and sulfate after the eruption of Mt Kelut in 2014. The results are new and of strong interest for scientist working on stratospheric sulfate and the impact of volcanic eruptions on climate. The paper describes an important new process. The readability of the paper is strongly improved. The text is now quite informative.

My major concern in my first review has been the injection area, raising the question that in case you can not get a comparable lifetime with heterogeneous reactions on ash and one grid box injection, your hypothesis might be wrong. The authors write now that they tested whether their hypothesis holds also with an injection into a grid box instead of a latitude band. Unfortunately, they have not include a figure to prove this into the answer to reviewers. I am willing to trust the authors in this point as my concern is more related to the transport of the model than to the ash processes.

Line 750: add "Figure S4 shows two model cases when we inject both SO₂ and ash into one grid box. Figure S4 shows that we need to tune the uptake efficiency from 3×10^{-3} to 8×10^{-4} to match the observation. Note that using a uptake efficiency of 8×10^{-4} and a saturation coverage of 3×10^{16} #/cm² is not the only way to match the data we discuss in Supplement 2."

Add Figure S4:

Figure S4. The volcanic SO₂ burden from satellite observations (symbols) and two model simulations (lines). The definitions of dashed lines and solid lines are the same as Figure 2. The black lines use the uptake efficiency of 3×10^{-3} and the saturation coverage of 3×10^{16} #/cm²; The red lines use the uptake efficiency of 8×10^{-4} and the saturation coverage of 3×10^{16} #/cm².

I recommend publication of the paper. The topic is important for better understanding the processes related to the simulation of volcanic eruptions in climate models.

L 244 - 246: I wonder were the problem is when you know the shape of the particle in laboratory studies and in the model. Please rewrite.

Line 192: modified to "These numbers will be useful to climate modelers simulating volcanic clouds and assuming spherical particles, but because we assumed the particles are spherical they are not directly comparable to laboratory studies which often measure the Brunauer, Emmett and

Teller (BET) surface area of volcanic ash samples. The surface area of real volcanic ash varies case by case and is expected to be larger than the spherical assumptions.”

Figures: The font size used in many figures is still quite small.

We changed the font sized for Figure 3, Figure 8, Figure 1, Figure S2, Figure S3, and Figure S9.

Figure S1: Use SI units hPa, not mbar.

We changed mbar to hPa for the whole manuscript.

Reference:

Goodman, A., Li, P., Usher, C., & Grassian, V. (2001). Heterogeneous uptake of sulfur dioxide on aluminum and magnesium oxide particles. *The Journal of Physical Chemistry A*, *105*(25), 6109–6120.

Kulkarni, D., & Wachs, I. E. (2002). Isopropanol oxidation by pure metal oxide catalysts: number of active surface sites and turnover frequencies. *Applied Catalysis A: General*, *237*(1), 121–137. [https://doi.org/10.1016/S0926-860X\(02\)00325-3](https://doi.org/10.1016/S0926-860X(02)00325-3)

Maters, E. C., Delmelle, P., Rossi, M. J., & Ayriss, P. M. (2017). Reactive uptake of sulfur dioxide and ozone on volcanic glass and ash at ambient temperature. *Journal of Geophysical Research: Atmospheres*, *122*(18), 10–077.

Urupina, D., Lasne, J., Romanias, M., Thiery, V., Dagsson-Waldhauserova, P., & Thevenet, F. (2019). Uptake and surface chemistry of SO₂ on natural volcanic dusts. *Atmospheric Environment*, *217*, 116942.

Usher, C. R., Al-Hosney, H., Carlos-Cuellar, S., & Grassian, V. H. (2002). A laboratory study of the heterogeneous uptake and oxidation of sulfur dioxide on mineral dust particles. *Journal of Geophysical Research: Atmospheres*, *107*(D23), ACH 16-1. <https://doi.org/10.1029/2002JD002051>

Reviewers' Comments:

Reviewer #1:

Remarks to the Author:

Thanks to the authors for addressing the outstanding issues from the previous round of review.

(i) The authors' argument that the very high saturation coverage on spherical ash would in fact be 1-2 orders of magnitude lower if normalised to SSA_{bet} instead of SSA_{geo} is reasonable, and at least the discrepancy is now acknowledged explicitly in the manuscript.

(ii) Please note that there are still a few instances where 'sulfuric acid' should be changed to 'H₂SO₄' for consistency, and that Figures 7 and S10 are missing x-axis labels. I also suggest replacing '#/cm²' with 'molecules/cm²' for clarity when expressing saturation coverage of SO₂ on ash.

I hope it helps but please feel free to let me know if you've any questions or concerns

Reviewer #2:

Remarks to the Author:

I recoment publication of the paper. The topic is important for the understanding of processes related to the simulation of volcanic eruptions in climate models.

Fig S4: You should add to the figure caption that you show results of an injection into one box. You may also add the corresponding lines of Figure 2 for better comparison.

Do you plan to show Figure 1 in the first part of the paper? It is a bit puzzling this way.

Thank you for your time and effort to review our paper during this uncertain time.

Reviewer #1 (Remarks to the Author):

Thanks to the authors for addressing the outstanding issues from the previous round of review.

(i) The authors' argument that the very high saturation coverage on spherical ash would in fact be 1-2 orders of magnitude lower if normalised to SS_{Abet} instead of SS_{Ageo} is reasonable, and at least the discrepancy is now acknowledged explicitly in the manuscript.

Thank you.

(ii) Please note that there are still a few instances where 'sulfuric acid' should be changed to 'H₂SO₄' for consistency, and that Figures 7 and S10 are missing x-axis labels. I also suggest replacing '#/cm²' with 'molecules/cm²' for clarity when expressing saturation coverage of SO₂ on ash.

We changed the “sulfuric acid particle” to “sulfate particle” for consistency.

We change “#/cm²” to “molecules/cm²” throughout the content.

I hope it helps but please feel free to let me know if you've any questions or concerns

Thank you for your time and effort to review our paper during this uncertain time.

Reviewer #2 (Remarks to the Author):

I recoment publication of the paper. The topic is important for the understanding of processes related to the simulation of volcanic eruptions in climate models.

Fig S4: You should add to the figure caption that you show results of an injection into one box. You may also add the corresponding lines of Figure 2 for better comparison.

Change the title of Figure S11 to “The SO₂ evolution when we inject both SO₂ and ash into one grid box”. Add the SO₂onAsh case for comparison.

Do you plan to show Figure 1 in the first part of the paper? It is a bit puzzling this way.

I will leave it to the editor for suggestions.